# Genomic variants affecting homoeologous gene expression dosage contribute to agronomic trait variation in allopolyploid wheat

Fei He [1,2,22], Wei Wang [1,3,22], William B. Rutter [1,4], Katherine W. Jordan [1,5], Jie Ren[1,6], Ellie Taagen[7], Noah DeWitt [8,9], Deepmala Sehgal [10], Sivakumar Sukumaran [10], Susanne Dreisigacker [10], Matthew Reynolds [10], Jyotirmoy Halder[11], Sunish Kumar Sehgal[11], Shuyu Liu [12], Jianli Chen[13], Allan Fritz[14], Jason Cook[15], Gina Brown-Guedira[8,9], Mike Pumphrey [16], Arron Carter [16], Mark Sorrells[7], Jorge Dubcovsky [17], Matthew J. Hayden [18,19], Alina Akhunova [1,6], Peter L. Morrell [20], Les Szabo[21], Matthew Rouse[21] & Eduard Akhunov [1,3✉]

Allopolyploidy greatly expands the range of possible regulatory interactions among functionally redundant homoeologous genes. However, connection between the emerging regulatory complexity and expression and phenotypic diversity in polyploid crops remains elusive. Here, we use diverse wheat accessions to map expression quantitative trait loci (eQTL) and evaluate their effects on the population-scale variation in homoeolog expression dosage. The relative contribution of *cis*- and *trans*-eQTL to homoeolog expression variation is strongly affected by both selection and demographic events. Though *trans*-acting effects play major role in expression regulation, the expression dosage of homoeologs is largely influenced by *cis*-acting variants, which appear to be subjected to selection. The frequency and expression of homoeologous gene alleles showing strong expression dosage bias are predictive of variation in yield-related traits, and have likely been impacted by breeding for increased productivity. Our study highlights the importance of genomic variants affecting homoeolog expression dosage in shaping agronomic phenotypes and points at their potential utility for improving yield in polyploid crops.

[1] Department of Plant Pathology, Kansas State University, Manhattan, KS, USA. [2] State Key Laboratory of Plant Cell and Chromosome Engineering, Institute of Genetics and Developmental Biology, Chinese Academy of Sciences, Beijing, China. [3] Wheat Genetic Resources Center, Kansas State University, Manhattan, KS, USA. [4] USDA-ARS, U.S. Vegetable Laboratory, Charleston, SC, USA. [5] USDA-ARS, Hard Winter Wheat Genetics Research Unit, Manhattan, KS, USA. [6] Integrated Genomics Facility, Kansas State University, Manhattan, KS, USA. [7] School of Integrative Plant Science, Cornell University, Ithaca, NY, USA. [8] Department of Crop and Soil Sciences, North Carolina State University, Raleigh, NC, USA. [9] USDA-ARS SAA, Plant Science Research, Raleigh, NC, USA. [10] International Maize and Wheat Improvement Center (CIMMYT), Texcoco, Mexico. [11] Department of Agronomy, Horticulture and Plant Science, South Dakota State University, Brookings, SD, USA. [12] Texas A&M AgriLife Research, Amarillo, TX, USA. [13] Department of Plant Sciences, University of Idaho, Aberdeen, ID, USA. [14] Department of Agronomy, Kansas State University, Manhattan, KS, USA. [15] Department of Plant Sciences & Plant Pathology, Montana State University, Bozeman, MT, USA. [16] Department of Crop and Soil Sciences, Washington State University, Pullman, WA, USA. [17] Department of Plant Sciences, University of California, Davis, CA, USA. [18] School of Applied Systems Biology, La Trobe University, Bundoora, VIC, Australia. [19] Agriculture Victoria, AgriBio, Centre for AgriBioscience, Bundoora, VIC, Australia. [20] Department of Agronomy and Plant Genetics, University of Minnesota, St. Paul, MN, USA. [21] USDA-ARS Cereal Disease Lab, St. Paul, MN, USA. [22] These authors contributed equally: Fei He, Wei Wang. ✉email: eakhunov@ksu.edu

Whole-genome duplications (WGD) can provide short-term evolutionary advantages[1,2] and likely played an important role in the origin of most crops[3,4]. Wheat experienced more than one round of WGD[5]. Compared to their diploid relatives, polyploids have broader geographic distribution[6]. This suggests the importance of WGD for improving the crops' fitness in new environments, a factor that was critical for the spread of agriculture around the world[1,6]. The success of polyploid crops can potentially be attributed to the adaptive genetic diversity contributed by ancestral species or generated de novo after WGD[2,6–9]. The redundant genetic nature of polyploid genomes could promote the accumulation of novel variants without detrimental effects on fitness, consistent with the observed mutational robustness of polyploids[1,2,8,10–12].

Regulatory variants affecting gene expression levels play an important role in adaptive evolution and variation in complex traits[13,14]. WGD greatly expands the range of possible *trans*-interactions for regulatory variants controlling the expression of redundant homoeologous genes from different genomes[15,16]. This contributes to substantial changes in the expression patterns of polyploids relative to their diploid ancestors[4,17–20]. The genomic distribution of these *trans*-acting variants and their role in the regulation of homoeologous (duplicates from distinct sub-genomes) genes in polyploid crops is not well understood, though a transcriptomic study of polyploid cotton has highlighted the importance of *trans*-regulatory evolution for domestication[4]. An aspect of regulation unique to WGD is that homoeologous regulatory proteins can now interact with the redundant regulatory elements in the duplicated genomes creating a more complex regulatory network. In addition, many regulatory variants linked with one of the homoeologs have the potential to alter a gene's

expression and change its dosage relative to other homoeologs (Fig. 1a). While previous genetic mapping studies in allopolyploid wheat identified several genes where regulatory variants associated with adaptive and domestication traits[21–24] also change the relative levels of homoeolog expression, the overall impact of genomic variants on the population-scale variation in the relative expression of homoeologs and their role in shaping adaptive traits in polyploid crops remains poorly understood.

Here, we perform expression quantitative trait loci (eQTL) analysis using geographically and genetically diverse allohexaploid wheat (genome formula AABBDD) accessions. The study use association mapping to identify *cis*- and *trans*-acting variants that explain the variance in homoeologous gene expression, with gene expression treated as a phenotype. We partition the genetic variation of gene expression traits using the approach of Yang et al.[25]. This approach can separate the genetic effects of portions of the genome on gene expression variance, and we use it to explore the relative contribution of variants from the three wheat genomes to population-scale transcript abundance variation from homoeologous genes. We show that the relative contribution of *cis*- and *trans*-acting variants to expression of homoeologous genes is affected by demographic events and selection. The relative expression dosage of homoeologous genes is primarily defined by the frequency of rare and common *cis*-regulatory variants whose accumulation is associated with biased homoeolog expression. The analyses of frequency, effect sizes, and levels of linkage disequilibrium between the *cis*-variants of homoeologous genes suggest that the relative homoeolog expression dosage is under selection. By investigating the distribution of *cis*- and *trans*-acting eQTL across genomic regions showing distinct epigenetic marks and chromatin architecture[26–28], we demonstrate that

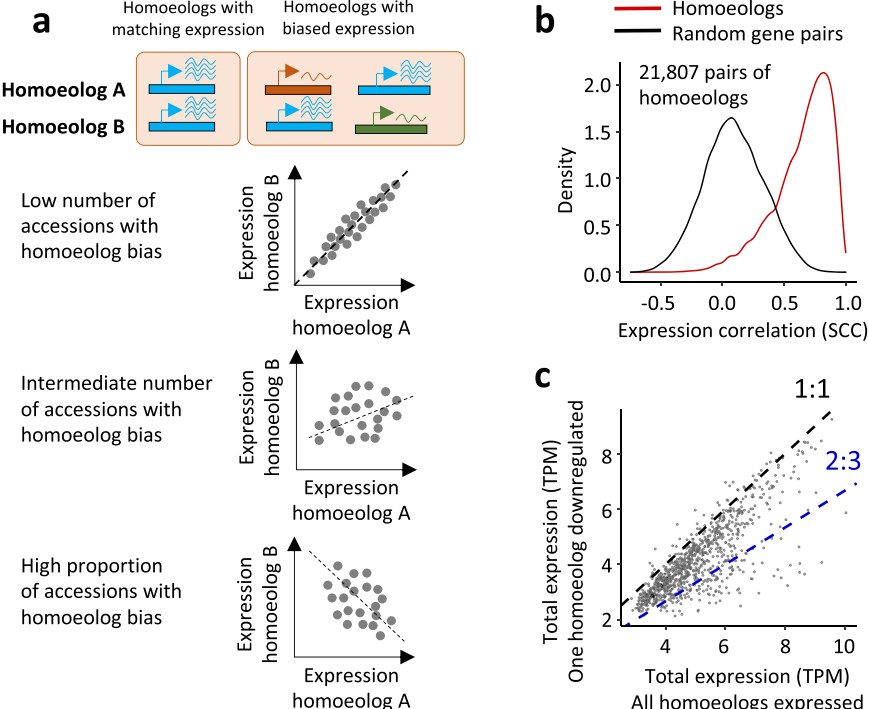

**Fig. 1 Relative expression of homoeologous genes in the diverse panel of wheat lines. a** Homoeologous gene pairs with matching and biased expression abundance of homoeolog A relative to homoeolog B. Red and green colors show low-expressing homoeologs in the A and B genomes, respectively. Increase in the frequency of accessions with a biased homoeolog is expected to reduce correlation between the levels of homoeolog expression measured in the panel. **b** Distribution of Spearman correlation coefficients (SCC) calculated between the AB, BD, AD gene pairs within the same homoeologous gene triplets (red) and random (black) pairs of genes using gene expression values from the 198 accessions. **c** The mean of the sum of the total triplet expression (A + B + D) in groups of accessions with (*y*-axis) and without (*x*-axis) one of the gene copies downregulated. The red and blue dotted lines show the 1:1 and 2:3 combined expression ratios, respectively. Source data are provided as a Source Data file.

eQTL are enriched in the regions of active chromatin. Finally, summary-level eQTL and GWAS mapping data[29], and gene co-expression networks (GCN) are analyzed jointly to study the role of variants linked with homoeologous gene regulation in shaping variation in major agronomic traits in wheat. We show that the frequency of genomic variants associated the relative expression dosage of homoeologous genes is predictive of variation in productivity traits in allopolyploid wheat and appear to be targeted by improvement selection. Thus, our study highlights the importance of WGD and emerging regulatory complexity in the evolution of phenotypic diversity that serves as a basis for the development of adapted crop varieties.

## Results

**Population-scale homoeologous gene expression variation**. We collected RNA-seq data from total RNA isolated from 2-week-old seedlings of 198 diverse accessions (Supplementary Data 1) selected to represent the broad geographic and genetic diversity of bread wheat. An average of 65.7 million paired-end Illumina reads ($2 \times 100$ bp) were collected for each sample, and after quality trimming mapped to the reference genome RefSeq v.1.0[5]. The proportion of reads unambiguously mapped to the individual wheat genomes was 81% (Supplementary Data 1). A simulation-based estimate suggested that the alignment settings used in our study provide 98% correct read mapping to the polyploid wheat genome (see "Methods"). Expression levels measured as Transcripts Per Million (TPM) were estimated for high-confidence (HC) genes in RefSeq v.1.0, with 52,511 transcripts (47,274 genes) showing TPM > 0.5 in at least three wheat lines (PRJNA670223) (Supplementary Data 2). In addition, we have analyzed RNA-seq previously generated for 90 wheat lines from spikes at the double-ridge development stage[30].

In allohexaploid wheat, genes appear as homoeologs present in three ('triplets') or two copies or as singletons[18]. Compared to singletons, on average, homoeologs in triplets showed higher expression levels (ANOVA $F$-test = 87, df = 1, $p = 2.2 \times 10^{-16}$), but lower expression variance (Supplementary Fig. 1a). The inter-genomic comparison of the population means of homoeolog expression in seedlings showed a positive correlation (Supplementary Fig. 1b), indicating that the relative expression levels of the most of homoeologs in our wheat panel tend to match. As a measure of the relative expression of homoeologs at the population level, we used Spearman Correlation Coefficient (SCC), which was calculated for each pair of homoeologs using their expression values in the panel of 198 accessions. While a strong positive correlation would be indicative of matching homoeolog expression levels (Fig. 1a) in most accessions in the panel, an increase in the proportion of accessions with biased homoeologs would decrease SCC (Fig. 1a). Compared to random pairs of genes selected from the distinct wheat genomes, the SCC distribution for the pairs of homoeologs was strongly shifted towards positive values (Fig. 1b, c), suggesting that the majority of accessions in the panel carry homoeologs with the matching levels of expression. The SCCs calculated for the same sets of homoeologs using RNA-seq data from both the seedlings and spike tissues[30] collected from a distinct set of accessions were generally similar, suggesting that tissue-specific factors do not substantially affect co-expression of the majority of homoeologs at the population scale (Supplementary Fig. 1c).

In polyploids, the relative dosage of duplicated genes tends to be balanced[18,31,32]. To investigate whether the downregulation of one of the homeologs in the population is compensated by increased expression of other homoeologs, we selected a set of 1443 gene triplets that met two criteria: (1) one out of three homoeologs was downregulated (TPM < 0.1) in at least two wheat lines, and (2) at least two wheat lines have all three homoeologs expressed (TPM > 2). We applied these criteria to each triplet to split 198 accessions into two groups, one group composed of accessions with one of the homoeologs downregulated and another group including accessions with all three homoeologs expressed. The sum of expression values from all three homoeologs (A + B + D) was calculated for each accession and used to derive the mean of total homoeologs' expression for each group. In most cases, the mean expression ratio between these two groups across gene triplets (Fig. 1c) was below 1:1 but above 2:3, suggesting that decreased combined expression associated with homoeolog downregulation is not fully compensated by increase in the expression of other homoeologs.

**Partitioning genetic variance for gene expression traits onto different wheat genomes**. The genetic architecture of gene expression could be complex and driven by multiple *cis*- and *trans*-acting variants with a broad range of effect sizes. To account for the cumulative effect of all SNPs from the distinct wheat genomes on the expression of individual homeologs, we performed partitioning of genetic variation (Fig. 2a)[25]. For this purpose, 2,021,936 SNPs with MAF > 0.05 identified in our panel were grouped into three genome-specific sets from the A, B, and D genomes. These sets were used to build genetic relationship matrices and estimate the genetic variance[25] for the expression of each gene (Supplementary Data 3). SNPs located within the same genome as a gene of interest were referred to as *cis*-genomic SNPs, whereas SNPs located in other genomes were referred to as *trans*-genomic SNPs (Fig. 2a). For the 10,000 most highly expressed genes, the mean of gene expression variance explained by the entire SNP set from all three wheat genomes was 40.4% (Supplementary Fig. 2a and 2b). The D genome explained a lower proportion of variance (7.7%) in gene expression than either the A (19.1%) or B (13.6%) genomes (Mann–Whitney test $W_{A/D} = 23{,}172{,}000$, $p$-value $< 2.2 \times 10^{-16}$; $W_{B/D} = 41{,}122{,}000$, $p$-value $< 2.2 \times 10^{-16}$) (Supplementary Figs. 2a and 2b).

We found that, on average, *cis*-genomic SNPs from the A or B genomes explained a higher proportion of gene expression variance (21.7% and 28.7%, respectively) than *trans*-genomic SNPs (5–17%) from other genomes (Fig. 2a). However, in the evolutionarily younger D genome[33], the proportion of variance explained by the *cis*-genomic SNPs (12%) was only slightly higher than that explained by the *trans*-genomic SNPs from the A (11%) and B (9%) genomes (Fig. 2a). The results of variance partitioning in the developing spikes[30] were consistent with the results obtained in seedlings (Fig. 2a).

The expression variance of a number of genes was largely explained by the *trans*-genomic SNPs (Supplementary Data 3). Among 34,691 genes with at least 20% of the total variance in gene expression explained jointly by SNPs from all three genomes, 6173 genes (17.8%) had <1% of variance explained by the *cis*-genomic SNPs, with the remaining variance explained by the *trans*-genomic SNPs. While this group of genes showed significantly reduced *cis*-regulatory diversity compared to *trans*-regulated genes (Fig. 2b), no significant reduction of diversity between *cis*- and *trans*-only regulated genes (Wilcoxon rank sum test p-value = 0.1) was found in wild emmer using data from the previously published study[8], indicating that shift towards *trans*-regulation in these genes from the A and B genomes is likely associated with diversity loss during wheat improvement. Of these 6173 genes, 47.8% were located in the D genome, which shows reduced diversity ($\pi_D = 0.0003$) relative to the A and B genomes ($\pi_A = 0.0007$ and $\pi_B = 0.0010$)[34]. This result also indicates that polyploidization bottleneck, which disproportionately affected the D genome, explains most of the *cis*-regulatory diversity loss in wheat.

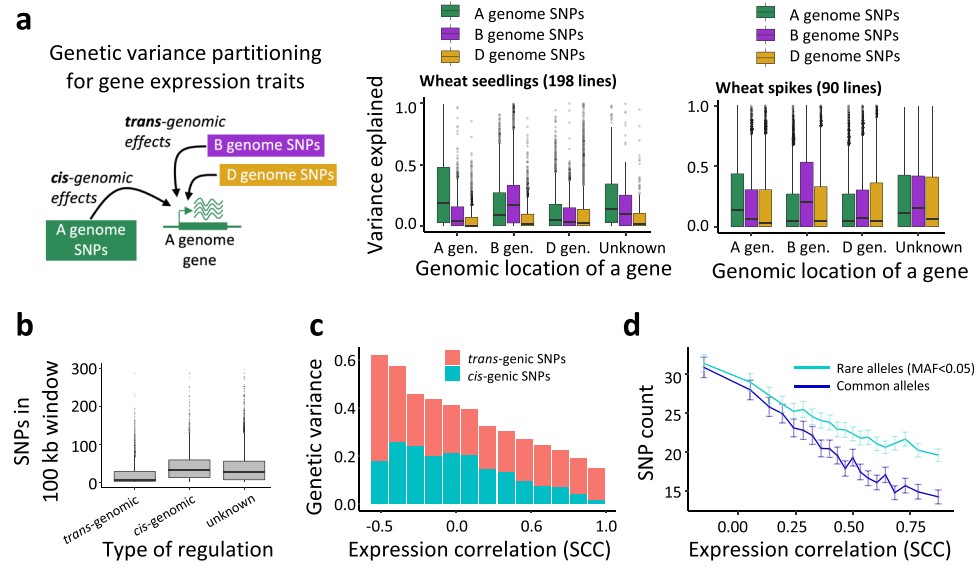

**Fig. 2 Partitioning variance in homoeolog expression using SNPs from different parts of the wheat genomes. a** An example of genetic variance partitioning for a gene located in the A genome using SNPs from the same genome (*cis*-genomic SNPs) or other homoeologous genomes (*trans*-genomic SNPs). As shown in the seedling panel, for genes in the A genome, variance explained by *cis*-genomic SNPs was 3.1 and 5.7 times higher than that explained by the B and D genomes' *trans*-genomic SNPs, respectively. Expression variance in the B genome was better explained by the B genome SNPs, which explained 1.3 and 3.4 times more variance than the SNPs from the A and D genomes, respectively. The variance explained by the *cis*-genomic SNPs from the D genome was comparable to that explained by the *trans*-genomic SNPs from the A and B genomes. Top 10,000 genes showing the highest expression variance were used in the analyses. **b** A 52.2% reduction in the mean SNP diversity (two-sided Mann–Whitney test $W = 142,90,644$, $p$-value $< 2.2 \times 10^{-16}$) was observed near 6173 genes with the expression variance mostly explained by *trans*-genomic SNPs, compared to 2852 genes with the expression variance explained predominantly by the *cis*-genomic SNPs. In (**a**) and (**b**), box shows the median and interquartile ranges (IQR). The end of the top line is the maximum or the third quartile (Q) + 1.5 × IQR. The end of the bottom line denotes either the minimum or the first Q − 1.5 × IQR. The dots are more or less than Q ± 1.5 × IQR. **c** The relationship between the proportion of genetic variance explained by *cis*- and *trans*-genic SNPs calculated for individual homoeologs and the levels of expression correlation (SCC) between the pairs of homoeologs in the wheat panel. The mean of genetic variance was calculated for data binned based on the SCC values. **d** The counts of rare and common SNPs in the genic regions of the 21,809 pair-wise combinations of homoeologs (gene body ± 10 kb) showing different levels of expression correlation. The mean and standard error of SNP counts were calculated for data binned based on the ranked SCC values. Source data are provided as a Source Data file.

We further split all SNPs into *cis*-genic (SNPs within ±1 Mb region around genes) and *trans*-genic (SNPs outside of ±5 Mb region around genes and SNPs located on other wheat chromosomes) subsets. The partitioning of variance for gene expression showed that the average variance explained by *trans*-acting variants (19.3%) is substantially higher (Wilcoxon test, $W = 3,228,268$, $p$-value $< 2.2e−16$) than that explained by *cis*-acting variants (12.3%). We also compared the means of total variance explained for two groups of genes: (1) genes with variance explained only by *cis*-genic SNP, where *trans*-genic SNPs contribute <1% to variance, and (2) genes with variance explained only *trans*-genic SNPs, where *cis*-genic SNPs contribute <1% to variance. The means of variance in these two groups of genes were similar (47.0% vs. 46.4%), suggesting that in the cases of *cis*-regulatory diversity loss in the allopolyploid genome, the contribution of *trans*-genic variants to expression variance is similar to the contribution of *cis*-genic variants.

**The effects of *cis*- and *trans*-acting variants on expression correlation between homoeologs.** The accumulation of rare and common mutations in gene promoters leads to dysregulation of gene expression in diploid genomes[35]. Even though polyploidy was expected to increase the mutation load in wheat[8,18] and result in expression dosage bias, the majority of homoeolog pairs in our study showed matching levels of expression. To better understand the genetic basis of homoeolog co-expression, we compared the proportions of expression variance in individual homoeologs explained by *cis*- and *trans*-genic SNPs among pairs showing the

distinct levels of expression correlation (SCC) (Fig. 2c). An increase in SCC was accompanied by a decrease in the total variance explained, with the largest proportion of explained variance observed for homoeologs with SCC < 0 (Fig. 2c). While SCC increase was accompanied by threefold decrease in variance explained by *trans*-genic SNPs, more substantial 11-fold decrease in variance explained was observed for *cis*-genic SNPs, reaching only 1.6% for homoeologs showing high correlation in the expression levels (SCC > 0.90) (Fig. 2c). These results suggest that discordant expression of homoeologs in the panel is likely associated with the accumulation of *cis*- rather than *trans*-regulatory diversity affecting the homoeologous genes. This conclusion is consistent with a decrease in the number of common (MAF ≥ 0.05) and rare (MAF < 0.05) SNPs around the homoeologs with an increase in levels of their expression correlation (Fig. 2d). These trends were consistent across all three wheat genomes (Supplementary Fig. 3a and 3b), indicating that negative relationship between *cis*-genic diversity and SCC is independent of the levels of genetic diversity in individual genomes. The lack of strong relationship between the inter-genomic sequence divergence and SCC suggests that divergence in the regulatory regions inherited from diploid ancestors unlikely has substantial global impact on the relative levels of homoeolog expression (Supplementary Fig. 3c).

To assess the impact of rare *cis*-variants (MAF < 0.05) on the homoeolog expression levels, we investigated the relationship between the expression ranks of each homoeolog in the population, ordered from lowest to highest expression levels across wheat accessions, and the rare allele load in the upstream

5-kb regulatory regions. In humans and maize[35,36], the total number of rare alleles in the regulatory regions coincides with the extremely low or high levels of expression in the population. We observed a similar trend using the entire set of homoeologous genes (Supplementary Fig. 2c). However, a subset of homoeologs showing high levels of expression correlation (SCC > 0.8) showed no enrichment for down- or upregulating rare alleles (Supplementary Fig. 2d). Combined together, our results suggest that the biased expression of homoeologs in the panel is primarily associated with the accumulation of both common and rare *cis*-acting variants.

**Mapping and functional annotation of variants affecting gene expression variation**. We conducted genetic mapping of *cis*- and *trans*-acting variants (eQTL) associated with the expression variation of individual genes among 191 wheat lines (Supplementary Data 1). After LD-based merging ($r^2 \geq 0.2$), we identified 36,898 and 15,238 significant SNPs (FDR < $10^{-5}$) in the RNA-seq datasets from wheat seedlings and spikes, respectively (Supplementary Data 4 and 5; Supplementary Figs. 4 and 5). A conservative criterion was applied to define *trans*-eQTL as eQTL located in different genomes or chromosomes relative to the target gene, and *cis*-eQTL as eQTL located ±1 Mb around a target gene. According to these criteria, in the seedlings, 8568 *cis*-eQTL and 14,645 *trans*-eQTL were associated with the expression of 8315 (8837 transcripts) and 8255 (8500 transcripts) genes, respectively. Out of these *cis*- and *trans*-eQTL in the seedlings, 247 eQTL affecting the expression of 1,469 genes (1500 transcripts) overlapped. In the RNA-seq data from spikes[30], we identified 3172 *cis*-eQTL for 3476 transcripts, and 9891 *trans*-eQTL for 7250 transcripts. The location of a *cis*-eQTL density peak averaged across all target genes relative to the coding sequence start site was similar between the A and B genomes but was ~200 bp more distant for the D genome (Supplementary Figs. 6a, 6b) and likely to be a consequence of lower diversity and more extended LD in the D genome[34].

The functional properties of identified eQTL were evaluated by calculating their enrichment within the specific regions of the genome tentatively affecting the coding potential of a gene or its regulation. The effects of SNPs on coding sequences were assessed using SNPeffect[37]. SNPs resulting in splice-site disruption and premature termination codons were considered as putatively deleterious. The regulatory regions were previously defined based on distinct epigenetic marks and open chromatin using a combination of MNase digest[26], DNaseI digest, and combined analyses of epigenetic variation, chromatin immunoprecipitation, and DNase-seq data[27]. The greatest levels of *cis*- and *trans*-eQTL enrichment relative to all variants in the genome were found for putatively deleterious variants, followed by missense and synonymous variants (Fig. 3a). Both *cis*- and *trans*-eQTL were found enriched in the regulatory regions (Fig. 3b, c) and depleted in the regions of closed chromatin hyper-resistant to MNase treatment (Fig. 3b). The *cis*- and *trans*-eQTL showed similar levels of enrichment across the various epigenetic marks, with both types of eQTL enriched for epigenetic modifications associated with gene body (H3K4me1), transcription (H3K36me3), and active expression (H3K27ac, H3K4me3) (Fig. 3e, f)[27,38,39]. Simultaneously, we observed a depletion of both *cis*- and *trans*-eQTL within epigenetic marks often associated with the repression of gene expression (H3K27me3) or transposable elements (H3K9me2) (Fig. 3c)[40,41].

Recent studies indicate that interaction between distant regulatory elements and their target genes could be facilitated by 3D chromatin contacts[42], which is consistent with the enrichment of *cis*-eQTL within regions involved in the formation

of chromatin loops in humans and maize[42,43]. To investigate the potential involvement of chromatin loops in gene expression regulation in wheat, we compared the distribution of eQTL-target gene pairs across the interacting regions identified by Hi-C[28] in cultivar Chinese Spring, which was not part of our diversity panel. Although, the low resolution of wheat Hi-C data does not allow us to map precisely regulatory regions involved in interaction, it could be used to assess the enrichment of eQTL-target gene pairs within chromatin loops relative to randomized data. First, we found that both *cis*- and *trans*-eQTL *p*-values positively correlate with the frequency of Hi-C contacts. This suggests that regions harboring eQTL-gene pairs showing stronger association are also more likely to have a higher frequency of chromatin contacts than regions harboring eQTL-gene pairs showing weaker associations (Fig. 3d, e, Supplementary Data 6). Second, the regions harboring *trans*-eQTL between both homoeologous and non-homoeologous chromosomes showed elevated Hi-C contacts (log10[Hi-C] = 1.24) compared to a distribution based on the 100 randomized samples (mean log10[Hi-C] = 0.92) (Fig. 3g). This result indicates that the probability of *trans*-eQTL-target gene pair occurrence within the chromatin loops is substantially higher than within the randomly selected regions. Among *trans*-eQTL-target gene pairs with a Hi-C contact frequency >50, 15% were located with the homoeologous chromosome regions, which are involved in chromatin interaction more frequently than non-syntenic regions (Fig. 3f, Supplementary Data 6)[28].

**Genetic architecture of homoeologous gene expression regulation**. To better understand the role of polyploidy in the regulation of homoeologous gene expression, we analyzed the genomic distribution of *trans*-eQTL and their gene targets. The total number of *trans*-eQTL in the A and B genomes for target genes in the same genomes was similar (Fig. 4a). However, the total number of *trans*-eQTL in the A and B genomes targeting genes in the D genome was 4.0 and 3.6 times higher than the total number of *trans*-eQTL in the D genome targeting genes in the A and B genomes, respectively (Fig. 4a). These results are consistent with the differences in the levels of genetic diversity between the wheat genomes[34] that also contributed to differences in the proportions of genetic variance for gene expression explained by SNPs from different genomes (Fig. 2a).

Also, we observed a tendency for the co-localization of *trans*-eQTL and target genes in the syntenic regions of homoeologous chromosomes (Fig. 4b, Supplementary Figs. 4 and 5). These patterns of *trans*-eQTL-target gene distribution are likely associated with the presence of shared regulatory elements in homoeologous genes that influence their co-regulation by regulatory feedback loops conserved among homeologs, as was demonstrated for the three homoeologs of the *Vrn-1* gene[44]. This hypothesis is supported the finding that, in a set of the 6,371 homoeologous gene triplets, 23% of homoeologs shared at least one eQTL, and that correlation in expression levels between the homoeologs increased with an increase in the proportion of shared eQTL (Fig. 4c).

Prior studies showed that gene expression is under purifying selection[45,46]. However, it remains unclear how genetic redundancy provided by polyploidy would affect selection on the expression of homoeologs. To answer this question, we compared the relationship between the minor allele frequency and the effect size of *cis*-eQTL for two groups of genes, homoeologs and singletons. Purifying selection acting against *cis*-eQTL with strong effects on gene expression results in negative correlation between *cis*-eQTL minor allele frequency and effect size[45]. We argue that if genetic redundancy influences the strength of purifying selection on the expression of duplicated genes, we expect to see the

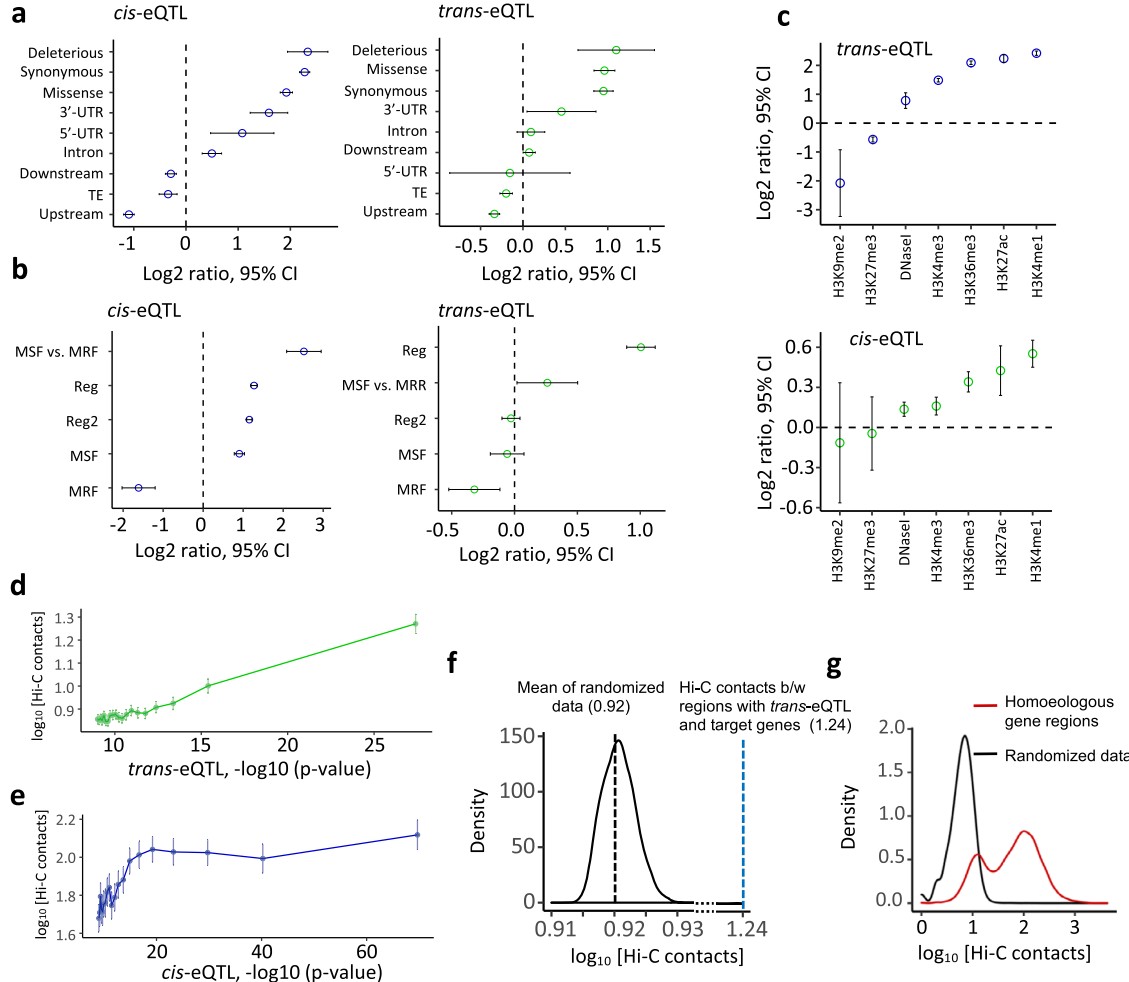

**Fig. 3 Functional annotation of eQTL.** The x-axis represents eQTL enrichment expressed as the $\log_2$ of the proportion of eQTL within specific classes of SNP variants (y-axis) relative to the proportion of eQTL within the random samples of SNPs. All analyses of enrichment and Hi-C contacts are based on 2,021,936 SNPs, 14,645 trans-eQTL and 8568 cis-eQTL. **a** Mean enrichment of cis-eQTL and trans-eQTL among SNP variants from different functional classes defined based on gene and transposable element (TE) annotation ($N = 2,021,936$ SNP sites). **b** Mean enrichment of cis-eQTL and trans-eQTL in the MNase Sensitive Footprints (MSF), MNase Resistant Footprints (MRF), and regulatory regions identified based on the sensitivity to DNase I treatment, epigenetic variation, and open chromatin marks. Enrichment was assessed relative to genome-wide patterns, except for MSF vs MRF, where enrichment was tested for eQTL located within MSF relative to MRF. 'Reg' region corresponds to regulatory elements, as defined in Li et al.[27]. 'Reg2' corresponds to regions classified as states 5–7 in the same study. **c** Mean enrichment of cis-eQTL and trans-eQTL in the epigenetically marked regions. **d** Relationship between the binned trans-eQTL p-values and frequency of Hi-C contacts between regions harboring a trans-eQTL and its target gene. Data are presented as mean values ± SEM. **e** Relationship between the binned distal (>2 Mbp) cis-eQTL-target gene p-values and frequency of Hi-C contacts between regions harboring a cis-eQTL and its target gene. Data are presented as mean values ± SEM. **f** Comparison of Hi-C contacts (log10-transformed) between regions harboring trans-eQTL-target gene pairs to Hi-C contacts between a random set of genes (two-sided Wilcoxon rank-sum test, $W = 191,789,322$, p-value = 8.8e−16). **g** Comparison of Hi-C contacts (log10-transformed) between the pairs of homoeologs and random pairs of genes.

reduced or lack of correlation between the allele frequency and effect size in homoeologs compared to singletons. However, we found that both groups of genes showed significant negative correlations (homoeologs: SCC = −0.23, $p < 2.2e{-}16$; singletons: SCC = −0.20, $p = 0.002$) between the cis-eQTL minor allele frequency and effect size (Fig. 4d). There was no significant difference between the two correlation coefficients (Fisher's z-test: $z = 0.53$, p-value = 0.60). The negative relationship between frequency and effect size was observed in homoeologs even for the subset of cis-eQTL whose effects are detectable at all frequencies (Supplementary Fig. 7). These results indicate that the expression levels of both homoeologs and single-copy genes are likely under purifying selection. Compared to homoeologs, the single-copy genes had cis-eQTL effect sizes higher across all MAF classes, likely due to the increased contribution of

trans-acting variants to the expression variation of homoeologs compared to that of the single-copy genes.

To understand the effects of cis- and trans-eQTL on the relative levels of homoeolog expression in our panel, we investigated the distribution of expression correlation values between the pairs of homoeologs (SCC) for the sets of homoeologs grouped based on the following eQTL-target gene configurations: (1) a homoeolog pair has no eQTL, (2) a homoeolog pair is regulated by shared trans-eQTL and has no cis-eQTL, (3) each gene within a homoeolog pair is regulated only by cis-eQTL, and (4) one of the genes within a homoeolog pair has cis-eQTL that also acts as a trans-eQTL for another homoeolog (Fig. 4e, Supplementary Data 7). We found that homoeolog pairs with no eQTL associated with their expression showed high levels of expression correlation (mean SCC = 0.57),

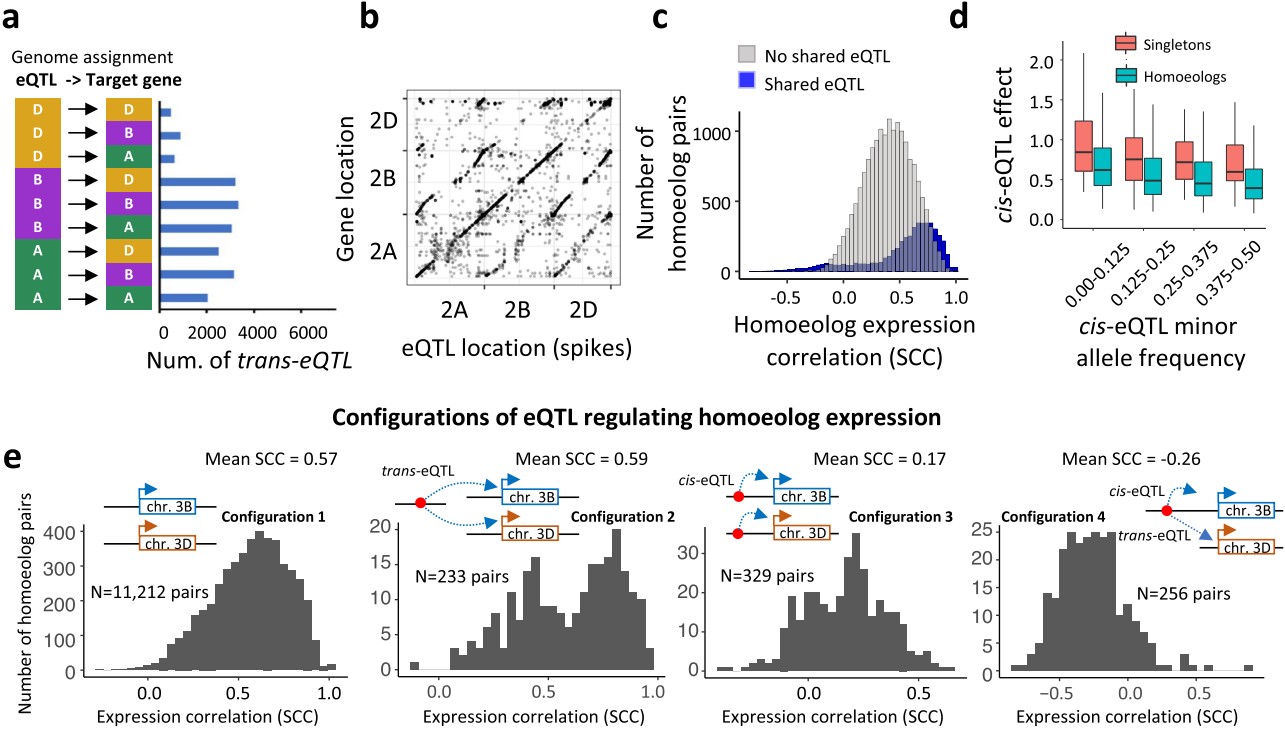

**Fig. 4 Effects of *cis*- and *trans*-eQTL identified in wheat seedlings and spikes on homoeologous gene expression. a** The total number of *trans*-eQTL targeting genes located either in the same genome (A → A, B → B, D → D) or in different genomes (A → B, A → D, B → A, B → D, D → A, D → B). **b** Location of eQTL relative to positions of target genes on wheat chromosomes 2A, 2B, and 2D in wheat spikes. **c**. Distribution of Spearman Correlation Coefficient (SCC) estimated for the pairs of homoeologs that either share at least one eQTL or do not share any eQTL. **d** The relationship between *cis*-eQTL minor allele frequency and effect size (absolute values) for two groups of genes, homoeologs, and singletons. Box shows the median and interquartile ranges (IQR). The end of the top line is the maximum or the third quartile (Q) + 1.5 × IQR. The end of the bottom line denotes either the minimum or the first Q − 1.5 × IQR. **e** Distribution of expression correlation values between the pairs of homoeologs for various eQTL - target gene configurations. A pair of homoeologs from chromosomes 3B and 3D was used to illustrate locations of *cis*- and *trans*-eQTL relative to their target genes. Source data are provided as a Source Data file.

consistent with the observed decrease in both SNP diversity (Fig. 2d) and expression variance explained by SNPs (Fig. 2c) with an increase in the homoeolog expression correlation. The expression correlation between homoeologs regulated only by shared *trans*-eQTL without effects detected from *cis*-eQTL showed bi-modal distribution with most homoeologs having a strong positive correlation (mean SCC = 0.59) (Fig. 4e); the subgroup with a higher correlation showed less diversity in the *cis*-regions than the subgroup with the lower correlations. These findings were consistent with the observation that the expression variance of the highly correlated homoeologs is defined largely by the *trans*-genic variants (Fig. 2c). Compared to these two eQTL-target gene configurations (1 and 2), more than a threefold reduction in the level of expression correlation between the homoeologs (SCC = 0.17) was associated with the presence of *cis*-eQTL, which appear to change the relative expression dosage of homoeologs in the panel. The set of homoeologs having variants acting as *cis*-eQTL for one homoeolog and *trans*-eQTL for another homoeolog showed a negative expression correlation (SCC = −0.26) (Fig. 4e). These cases represent an extreme form of homoeolog expression bias, where a high-expressing homoeolog in one genome is often associated with a low-expressing homoeolog on another genome (Fig. 1a).

**Joint eQTL and GWAS analyses of agronomic traits**. Recent studies have demonstrated the utility of gene expression data for interpreting GWAS results and identifying candidate causal genes by jointly analyzing eQTL and SNPs linked with trait

variation[29,43,47–49]. Consistent with these findings, we found a significant enrichment of *cis*-eQTL among SNPs associated with variation in yield component and development traits in a diverse set of wheat lines (Supplementary Fig. 8, Supplementary Data 8–9). In addition, we used the results of QTL mapping in bi-parental populations and diversity panels to assess the overlap of eQTL detected in our study with significant marker-trait associations (MTAs) for a number of agronomic traits identified in WheatCAP (www.triticeaecap.org) and IWYP (iwyp.org) projects (Supplementary Note 1, Supplementary Data 10). Using strict criteria for overlap (±1 kb), out of 1,112 non-redundant MTAs, 70 and 36 MTAs had *cis*-eQTL and *trans*-eQTL located within ±1 kb, respectively. While for *trans*-eQTL, this overlap was not substantially different from the randomized control, for *cis*-eQTL, this overlap was nearly two times higher than the maximum overlap of 33 eQTL obtained in the randomized control (Supplementary Fig. 8d and 8e). Consistent with an earlier study, these results suggest that the trait-associated SNPs are more likely to be *cis*-regulatory rather than *trans*-regulatory variants[50].

Further, we used GWAS and eQTL summary-level Mendelian Randomization (SMR) data analysis[29] to detect candidate genes whose expression levels co-vary with phenotypes due to pleiotropy or causal association (Fig. 5a). We obtained the SNP effects by performing GWAS for 14 productivity traits evaluated in two wheat populations, one characterized in this current study (see "Methods") and another characterized in the 1000 wheat exomes project[8]. By applying SMR to test for association between gene expression and productivity traits, we detected 971 and 424

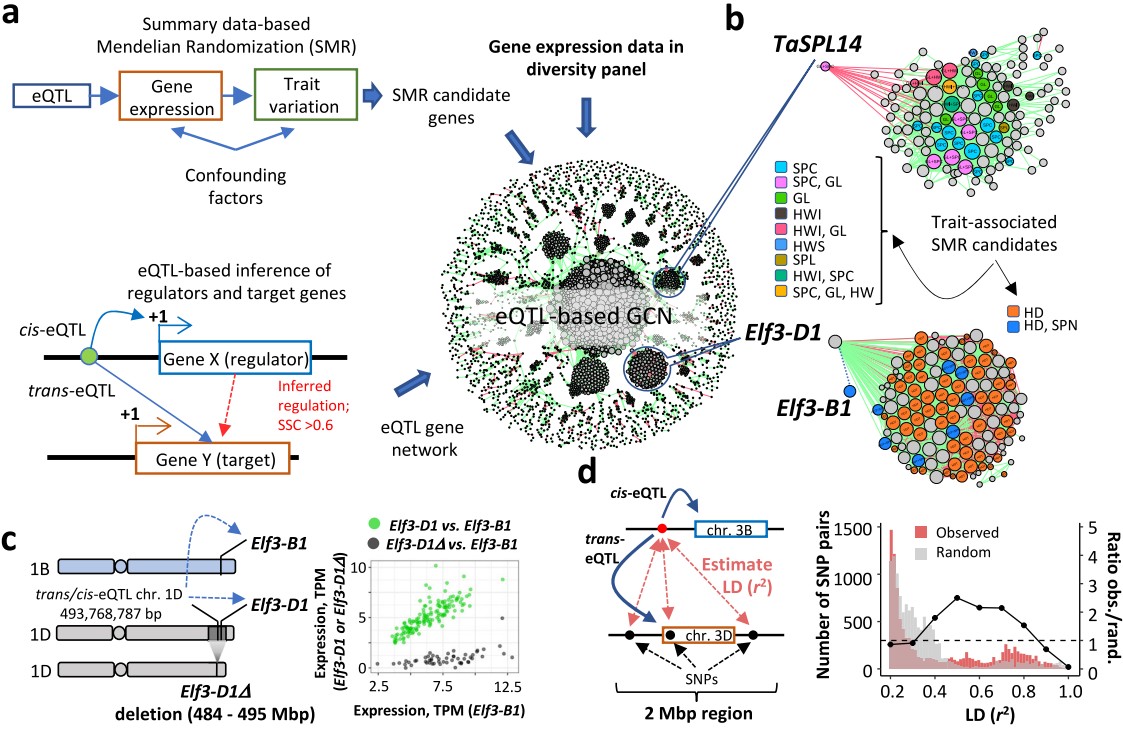

**Fig. 5 Joint eQTL and GWAS analysis of agronomic traits in wheat. a** Outline of the strategy used to integrate eQTL and GWAS data to investigate the genetic basis of yield component and development trait variation in wheat (see "Methods"). We used field-based phenotyping data collected for a diverse panel of ~800 wheat accessions from the 1000 wheat exomes project[8] including grain filling period (GFP), harvest weight (HW), drought susceptibility index for harvest weight (HWS), heading date (HD) and plant height (PHT) traits. A set of phenotypic traits was collected for a diverse panel of 400 wheat accessions: grain length (GL), grain width (GW), thousand-grain weight (TGW), grain area (GRA), spike compactness (SPC), spikelet number per spike (SPN), awnedness (AWN), and height (PHT) (Supplementary Data 10). **b** Gene co-expression network (GCN) modules, including *TaSPL14* and *Elf3* genes, are enriched for genes associated with agronomic traits in SMR analyses. **c** eQTL located on chr. 1D acts as a *cis*-variant for *Elf3-D1:TraesCS1D01G451200* (GWAS FDR-corrected $p$-value $= 4e-54$) and is tightly linked ($r^2 > 0.8$) with variants acting as *trans*-variants for *Elf3-B1:TraesCS1B01G477400* (GWAS FDR-corrected $p$-value $= 9e-10$). A deletion of *Elf3-D1* (*Elf3-D1Δ* locus) in wheat affects both HD and SPN traits. **d**. Distribution of LD between negatively correlated homoeologs compared to LD between the random set of homoeolog pairs. LD was measured between the *cis/trans*-eQTL in one homoeologue and SNPs within a 2-Mb window, including another homoeologue. Only LD values above $r^2 > 0.2$ were plotted. The right $y$-axis shows the ratio of SNP pairs within different LD ranges (the dotted line shows a ratio of 1.0) estimated for the negatively correlated and random pairs of homoeologs. Source data are provided as a Source Data file.

genes ($p$-value $< 10^{-4}$) using eQTL from the seedlings and spikes, respectively (Supplementary Data 11). Since the association of variants with gene expression and traits could be due to either pleiotropy or linkage, a HEIDI method was applied to distinguish between these two scenarios[29]. A total of 329 and 95 genes identified using the seedling and spike eQTL, respectively, passed the HEIDI test ($p$-value $\geq 0.05$) (Supplementary Data 11). Among these genes we had homologs that showed association with productivity and development traits in wheat and other plants, supporting the utility of SMR for investigating genetic mechanisms underlying trait variation in wheat. To connect the SMR candidate genes and biological pathways, we built an eQTL-based GCN (Fig. 5a), which includes genes co-expressed with the SMR candidate genes, as well as genes identified as regulators and regulatory targets in the wheat eQTL map (see "Methods", Supplementary Figs. 9 and 10, Supplementary Data 12–14).

In rice, *SPL14* was linked with increased panicle branching and yield[51], but no *TaSPL14* alleles positively affecting productivity traits in wheat were reported. Here, we show that the *TaSPL14* transcription factor (TraesCS5B01G512800) is associated with variation in spike compactness, grain length, and harvest weight (Fig. 5b, Supplementary Data 11), consistent with the decreased spikelet number and thousand-grain weight observed in wheat mutants with the *TaSPL14* gene knocked-out[52]. Among genes identified by SMR and connected with the *TaSPL14* in the eQTL

network (Fig. 5a, b) was the homolog of the *FAR1* gene, responsible for phytochome A-mediated far-red response, associated with flowering time regulation in *Arabidopsis* and wheat[53,54]. The wheat homologs of *FAR1* were also differentially expressed in wheat lines with the knocked-out homolog of *SPL14*[52], suggesting that *FAR1* is likely one of the downstream *TaSPL14* regulation targets in wheat.

The SMR analysis linked the Early Flowering 3 (*TaElf3*) gene in the B genome (*TaElf3-B1*) with heading date (HD) and spikelet number per spike (SPN) in our population (Fig. 5b). A subtelomeric deletion polymorphism, including the *TaElf3-D1* gene, also detected in our population (Supplementary Fig. 11a), was previously associated with the heading date variation in wheat[55]. We showed that *TaElf3-B1* expression is associated with a *trans*-eQTL on chromosome 1D (pos. 493,768,787 bp), which is in strong LD ($r^2 > 0.8$) with variants that act as a *cis*-eQTL for *TaElf3-D1* (Fig. 5c), and likely linked with the presence/absence variation (PAV) affecting *TaElf3-D1*.

The lack of the *TaElf3-D1* transcripts in accessions with the terminal deletion leads to biased expression of the *TaElf3-D1* and *TaElf3-B1* homoeologs, and negative correlation between the expression levels of these homoeologs in the panel (SCC $= -0.18$, $p$-value $= 0.01$) (Fig. 5c). The higher expression of *TaElf3-B1* in lines with the *TaElf3-D1* deletion than in the lines without the deletion ($t$-test $p$-value $= 2 \times 10^{-4}$) (Supplementary Fig. 11b)

suggests that the level of *TaElf3-B1* expression could be conditioned by the allelic state of the *TaElf3-D1* homoeolog. This result combined with the presence of 30 SNPs near *TaElf3-B1* showing high LD ($r^2 > 0.4$) with the *trans*-eQTL on 1D indicates that the combinations of the *TaElf3-D1* and *TaElf3-B1* alleles in our panel could be non-random.

**Allelic combinations of homoeologs showing negative expression correlation in the panel is not random.** Previous studies indicate that the non-random combinations of alleles showing elevated inter-locus LD could arise due to population structure, selection, or epistasis[56–59]. In our study, we identified a number of homoeologous gene pairs whose expression levels, similar to the *TaElf3-D1* and *TaElf3-B1* homoeologs, show a negative correlation in our panel, which results from the presence of accessions carrying homoeologs showing both matching and biased expression levels in the panel (Fig. 1a). To test whether negative expression correlation between homoeologs is associated with the non-random combinations of homoeologous alleles, we estimated inter-chromosomal LD between SNPs located near the pairs of negatively correlated homoeologs showing SCC < −0.4 (59 homoeologs in total). LD was calculated between the *cis*-eQTL associated with variation in the expression of one of the homoeologs and SNPs located within a 2-Mb region around another homoeolog (Fig. 5d). To take into account the effects of population structure, which could also lead to elevated inter-locus LD[57], we calculated LD between SNPs from the randomly selected pairs of homoeologs (Fig. 5d). Compared to this random set of SNPs, the regions harboring the negatively correlated homoeologs still showed nearly a two-fold increase in the proportion of high-LD SNPs ($r^2 > 0.4$) (Fig. 5d), suggesting that the allelic combinations of negatively correlated homoeologs in the panel could result from selection.

**Accumulation of homoeologs with biased expression affects agronomic traits.** Prior studies suggest that some adaptive traits in wheat could be impacted by the relative expression of the homoeologous copies of causal genes[21–24]. To better understand the overall impact of the biased homoeolog expression on phenotype, we investigated the relationship between productivity traits and the number of homoeologous alleles resulting in biased expression in the set of 59 negatively correlated homoeologs (SCC < −0.4), which were identified in the RNA-seq data from the seedlings (Supplementary Data 15). In the RNA-seq data from spikes, these homoeologs also showed the lack of coordinated expression (mean SCC = 0.03 ± 0.05), although not as substantial as in the seedlings. The majority of these homoeologs had low-expressing alleles in at least two wheat lines. On average, for this set of homoeologs, we detected eight low-expressing alleles per line, ranging from 1 to 31 per line in the panel (Fig. 6a–c, Supplementary Data 15). The minor allele frequency (MAF) of *cis*-eQTL associated with these negatively correlated homoeologs was shifted towards common variants, with the mean MAF of 0.30 ± 0.01 (Supplementary Fig. 12), indicating that the homoeologous alleles contributing to biased expression are present at high frequency in our panel. To assess whether the low-expressing alleles of homoeologs are associated with gene deletions, similar to the deletion of the *TaElf3-D1* homoeolog (Fig. 5c), we compared the sequences of homoeologs with the wheat PanGenome[60]. This analysis showed that the lack of transcripts from only two of 59 homoeologs could be linked with the presence/absence variation (Supplementary Data 16).

The majority of analyzed traits showed a significant positive or negative correlation with the total number of low-expressing alleles of homoeologs per line (Supplementary Fig. 13a), with the absolute correlation coefficients being higher than those obtained using the random control (Supplementary Fig. 13b). The accumulation of the low-expressing alleles of these homoeologs was associated with an increase in grain length (SCC = 0.26), width (SCC = 0.41) and weight (SCC = 0.39), and a decrease in heading date (SCC = −0.29), number of spikelets per spike (SCC = −0.35), spike length (SCC = −0.19), and plant height (SCC = −0.18) (Supplementary Fig. 13a). A similar analysis performed using the negatively correlated homoeologs detected in the spikes and the number of grains and the number of spikelets per spike also revealed negative correlation between the low-expressing homoeologous alleles and traits (SCC = −0.25 and SCC = −0.16, respectively).

We also tested for association between the negatively correlated homoeologs and phenotypes (Fig. 6a–c) by predicting productivity traits using the expression values of homoeologous alleles and ridge regression modeling[35]. Except for spike length, spike compactness, and awnlessness traits, the correlation between predicted and observed trait values for major productivity traits ranged from 0.25 to 0.37 (Fig. 6d, Supplementary Table 1). For many traits, including spikelet number per spike and grain length, the correlation between the true traits and traits predicted using the expression levels of negatively correlated homoeologs was in the 99th percentile of distribution generated using expression data from the random sets of genes (Fig. 6e, Supplementary Table 1). This further confirms that the negatively correlated homoeologs are predictive of variation in productivity traits. These results combined with the observed correlation between the number of low-expressing alleles and the grain size/grain number traits (Supplementary Fig. 13b) suggest that the negatively correlated homoeologs could be connected with processes affecting variation in and trade-off between the productivity traits.

Association between the negatively correlated homoeologs and phenotype was independently validated using a panel of lines from the 1000 wheat exomes project[8]. By correlating the number of *cis*-eQTL alleles associated with the low-expressing alleles in the set of 59 negatively correlated homoeologs and traits we showed that an increase in the number of these alleles is linked with an increase in grain yield and decrease in heading date (Supplementary Fig. 14). Combined together, our results indicate that the accumulation of eQTL variants linked with the relative changes in the homoeolog expression dosage have potential to affect traits of agronomic importance in polyploid wheat.

## Discussion

We characterized the genetic variants associated with variation in homoeologous gene expression measured in a panel of diverse allopolyploid wheat lines. The enrichment of detected variants in the regions of active chromatin[26,27] suggests we uncovered many SNPs involved in regulatory function in the wheat genome. We used the developed eQTL resource to interpret GWAS results for complex productivity traits, which are subjected to human-driven selection during wheat improvement. Using extensive trait mapping data from the WheatCAP (www.triticeaecap.org) and IWYP (iwyp.org) projects, we showed significant enrichment of *cis*-eQTL around the top marker-trait associations. Applying a transcriptome-wide SMR analysis[29], we identified a number of candidate genes whose expression is linked with variation in these traits. We showed that *TaSPL14*, the rice ortholog of transcription factor *SPL14 (IPA1)* controlling plant architecture[51], is associated with natural variation in spike and spikelet development traits in wheat[51,52]. Our results suggest that joint modeling of GWAS and eQTL data using summary-level statistics has the potential to identify causal genes associated with trait variation in wheat or to prioritize candidate genes for further functional validation.

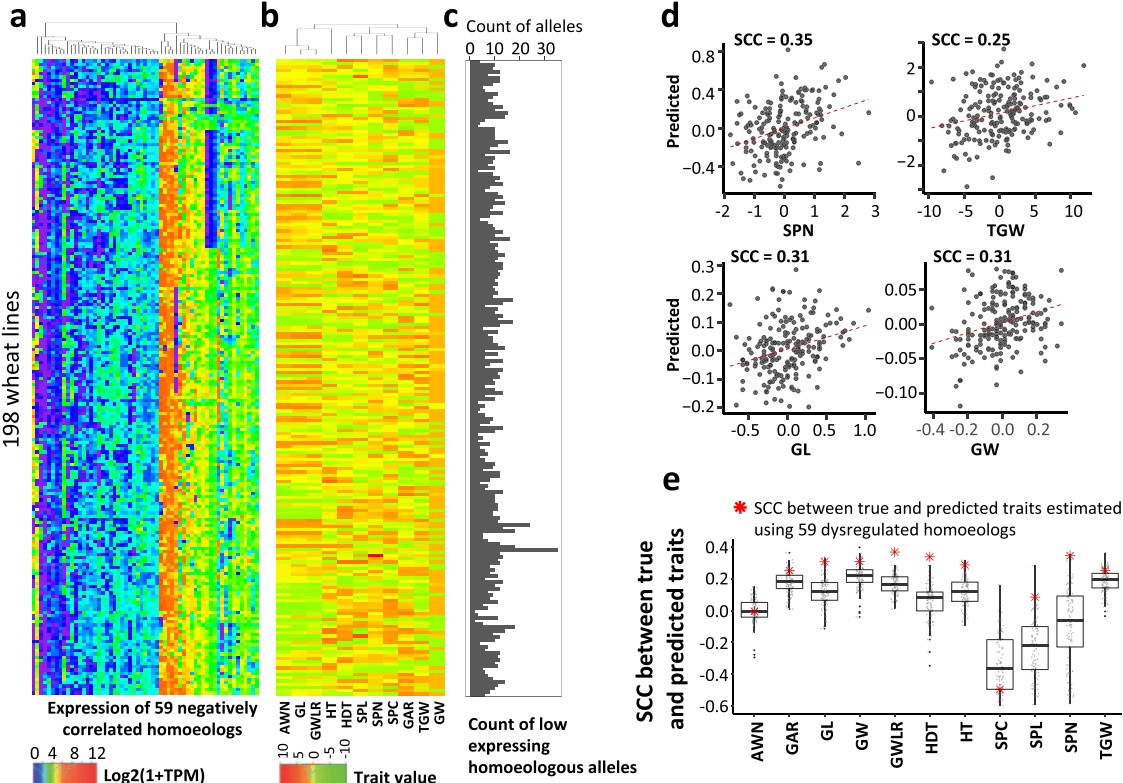

**Fig. 6 Biased expression of homoeologous genes is linked with variation in productivity traits. a** Hierarchical clustering of 198 wheat lines based on the levels of expression ($\log_2(1 + \mathrm{TPM})$) of 59 negatively correlated homoeologs. **b** Hierarchical clustering of 198 wheat lines based on the normalized productivity trait values. **c** The count of low-expressing alleles of homoeologs per line showing evidence of negative expression correlation in the population. **d** Correlation between the observed and predicted values for productivity traits. Predictions were performed using ridge regression based on expression data from the negatively correlated homoeologs. **e** Distribution of SCC between true and predicted traits from 100 replications of 10-fold cross-validation using the random sets of homoeologous genes and ridge regression modeling. Predictions generated using real-world data from negatively correlated homoeologs are shown by a red asterisk. Box shows the median and IQR. The end of the top line is the maximum or the third quartile (Q) + 1.5 × IQR. The end of the bottom line denotes either the minimum or the first Q − 1.5 × IQR. The dots are more or less than Q ± 1.5 × IQR. Source data are provided as a Source Data file.

The relative contribution of *cis*- and *trans*-acting variants to population-scale variation in homoeologous gene expression and relative expression dosage appears to be strongly influenced by the demographic events accompanying wheat origin (polyploidization and gene flow) and human-mediated selection for domestication and improvement traits[8,33,34,61]. The A and B genomes of hexaploid wheat are more genetically diverse than the D genome because of the post-polyploidization gene flow from tetraploid (AB genome) into hexaploid wheat[8,62–64]. This factor contributed to a higher number of *trans*-acting variants in the A and B genomes compared to that in the D genome and likely accounts for the similar proportion of expression variance explained by *cis*- and *trans*-genomic variants for genes in the D genome. The latter is in contrast to the higher proportions of gene expression variance explained by *cis*- rather than *trans*-genomic variants in the genetically more diverse A and B genomes.

Our study highlights in importance of inter-genomic *trans*-effects in the regulation of genes that lost their *cis*-regulatory diversity in polyploid wheat due to either selection or polyploidization bottleneck[8,60,63,65–67]. Among genes whose expression is largely explained by *trans*-genomic effects and that show the evidence of reduced genetic diversity, half were located in the D genome indicating that the loss of *cis*-acting diversity in these genes occurred during hybridization between the tetraploid wheat and the D genome ancestor[8]. The loss of *cis*-acting variants in the A and B genomes is likely associated with domestication and improvement selective sweeps, which affected a significant

portion of the wheat genome[8,60,65,66]. The cumulative effect of *trans*-acting variants on the expression variance of genes that lost *cis*-regulatory diversity was comparable to the effects *cis*-acting variants on genes showing no evidence of *trans*-regulation, indicating that inter-genomic interactions should play significant role in regulating genes controlling domestication and improvement traits in wheat.

We showed that in allopolyploid wheat, the relative levels of homoeolog expression are defined by the combination of *cis*- and *trans*-acting variants from all three genomes. On average, expression variance explained by *trans*-acting variants for all analyzed genes was 57% higher than variance explained by *cis*-acting variants. For homoeologous genes, the relative contribution of *cis*- and *trans*-acting variants for negatively correlated homoeologs (SCC < 0) was also comparable. However, with increase in the levels of homoeolog expression correlation contribution of *cis*-acting variants to expression variance significantly declined, indicating that these variants play more prominent role in creating expression dosage imbalance than *trans*-acting variants. The functional importance of *cis*-regulatory variants leading to homoeolog expression bias was confirmed by earlier studies, which showed that causal variants underlying several adaptive and domestication traits[21–24] also lead to changes in the homoeolog expression dosage.

Previous study in maize[35] linked the loss of fitness with gene expression dysregulation caused by rare mutations in the promoters of thousands of non-redundant genes. One of the

expected consequences of functional redundancy provided by polyploidy is increase in the mutation load in wheat[8,18] that could disrupt the co-expression of homoeologous genes by creating low- or high-expressing alleles. The prevalence of homoeologs with the matching levels of expression in our study could be explained by the recent origin of allopolyploid wheat[33], which provided less time for the accumulation of dysregulating mutations. However, our results also suggest that the expression levels of homoeologous genes could be under purifying selection, consistent with the results previously reported for diploid organisms[45,46]. The high levels of LD between the combinations of the high- and low-expressing homoeologous alleles showing negative expression correlation in our wheat panel, among others including the *Elf3* homoeologs, provides additional support for selection acting to maintain certain levels of homoeolog expression dosage. The *trans*-acting variants appear could provide some level of robustness against the dysregulating effects of *cis*-acting variants, as evidenced by the proportional increase in the homoeolog expression levels with an increase in the number of shared *trans*-eQTL. Nevertheless, this mutational robustness provided by the recent WGD in wheat[8,10] is not sufficient to fully compensate homoeolog expression bias by increasing the expression levels of corresponding homoeologs from other genomes. If this expression imbalance affects the regulatory pathways controlling adaptive traits, the homeolog expression dosage could be targeted by selection. This is in agreement with the previously reported evidence of deleterious variant removal from homoeologous genes in wheat[8].

Our study uncovered an association between the number of the common low-expressing alleles from homoeologs showing negative expression correlation and variation in productivity traits exhibiting trade-offs in wheat. A decrease in the total number of low-expressing alleles per line was accompanied by an increase in spikelet number per spike and a decrease in grain size and weight. Compared to a randomized control, the total number of low-expressing homoeologous alleles was more predictive of productivity trait variation, likely due to some connection of these homoeologs with trait-associated biological pathways. Recent yield increases in elite winter wheat cultivars were mostly linked with an increase in the number of spikelets and grains per spike[68,69], whereas in some cultivars from Asia, yield increase was mostly associated with an increase in grain size and weight[70]. These trends are in agreement with studies suggesting that the contribution of different productivity traits to increased yield potential is environment-specific[71]. Based on our results, we hypothesize that the low-expressing alleles of homoeologs creating dosage imbalance within the homoeologous gene sets, due to their impact on productivity traits in wheat, were targeted by improvement selection. Depending on which productivity trait was chosen as a breeding target, these homoeologous alleles were likely either purged (increases the number of spikelets/grains per spike) or accumulated (increases grain size/weight) in wheat lines. Identification of these homoeologous genes with imbalanced expression and associated pathways provides opportunities for targeted breeding or genome-editing strategies aimed at adjusting proportions of these alleles in the genome to maximize crop productivity.

## Methods

**Plant material**. A panel of 400 diverse wheat lines (Supplementary Data 1) was selected from a larger worldwide sample of 2259 *Triticum aestivum* accessions that were previously genotyped using the 9 K iSelect SNP array[72]. The seeds could be requested from the USDA National Small Grains Collection. Our panel was assembled to maximize: (1) genetic diversity, (2) representation of diverse geographic regions, and (3) representation of phenotypic response to the strains of fungal pathogen *Puccinia graminis* f. sp. *tritici* (*Pgt*). The panel of 2259 lines was previously evaluated in the Wheat CAP project by infecting plants at the seedling

stage using *Pgt* races TTKSK (Ug99), TRTTF, TTTTF, BCCBC, and a bulk of six races (TPMKC, RKRQC, RCRSC, QTHJC, QFCSC, and MCCFC). The phenotyping data is available from the Wheat CAP T3 database (https://triticaetoolbox.org). The Stakman infection types recorded on a 0–4 scale were converted to A through F grades where A corresponded to infection types '0' to ';1'; B to ';13' to '31;' mesothetic infection types; C to '2-' to '2'; D to '2+' to '32+'; and F to '3' to '4'. We selected 50 wheat lines that showed an 'F' grade to all five *Pgt* races. We also selected 350 additional lines that showed variable, race-specific responses to the *Pgt* races. When possible, 10 lines for a given pattern of infection type grades were selected. No lines possessed an A grade in response to all five races. During the selection of the lines within a given pattern of infection type grades, both geographic origin and PCs were used to maximize the diversity of the panel. A subset of 204 wheat lines representing geographic and phenotypic diversity of this diversity panel was subjected to RNA-seq analysis and used for the genetic dissection of gene expression variation traits. The genetic relatedness analysis of this subset of lines was performed using an algorithm implemented in PLINK v.1.9. For this purpose, we have used genome-wide SNPs generated by the regulatory sequence capture and sequence-based genotyping approaches. This analysis shows that our panel does not contain highly related accessions, which otherwise might increase the chances of detecting spurious associations in GWAS (Supplementary Fig. 15).

**RNA-seq data analysis**. Total RNA was isolated from 2-week-old seedlings of 204 lines, with each line grown in three biological replicates. Ground tissues from three biological replicates were combined in equal amounts before RNA isolation using the RNeasy Plant mini kit. RNA-seq libraries were prepared with TruSeq™ RNA Sample Prep Kit (Illumina) using the Beckman's Biomek® FXP Laboratory Automation Workstation. Up to eight barcoded RNA-seq libraries were pooled per lane of NextSeq2000 flow cell to generate 2 × 100 bp reads.

A total of 13,415,679,980 paired-end 2 × 100 bp reads were generated for 204 wheat accessions from the wheat diversity panel, with a mean of 65,763,137 reads per accession (GSE167479). The reads were mapped to the wheat RefSeq v.1.0 using HISAT2 (ver. 2.1.0) with the following parameters: --max-intronlen 70000, --dta. On average, 81% of all reads were mapped to the genome uniquely, with an average of 7% reads failing to map (Supplementary Data 1).

In addition, we have analyzed previously published RNA-seq data generated for 90 wheat lines from spikes at the double-ridge development stage[30]. Fastq files were downloaded from NCBI BioProject PRJNA348655 using 'fastq-dump' from the SRA Toolkit (v. 2.9.6). The spike RNA-seq dataset contained 46,394,170 paired-end 2 × 125 bp reads, of which 86% reads could be mapped to the reference genome uniquely, and 5% of reads failed to map.

We generated RNA-seq data for 2-week-old seedlings from 204 wheat lines. We removed samples with a substantial amount of rRNA contamination and samples with <40% uniquely mapped reads. The resulting set of 198 RNA-seq samples was used for further analysis (Supplementary Data 1). We used the Kallisto program that uses pseudoalignment of RNA-seq reads to reference gene models to assess the transcript abundance[73]. Its performance has previously been evaluated in the polyploid wheat genome[18].

All high confidence (HC) and low confidence (LC) gene models from the IWGSC RefSeq v. 1.0[5] were combined for estimating the TPM values using Kallisto (v. 0.4.6.0)[73]. Transcripts with expression standard deviation >0.5 and expressed (TPM > 0.5) in at least three samples have been used in our analyses. This set of included 52,511 transcripts from the HC gene models and 29,226 transcripts from the LC gene models. Only HC gene expression data were used for further analyses. The expression data were log2-transformed followed by robust quantile normalization in R. The probabilistic estimation of expression residuals (PEER) was used to remove hidden confounding factors in the expression data[74], and residuals were used for studying the genetic effects on expression levels in the population.

To assess the accuracy of transcript abundance estimation by mapping RNA-seq reads to the polyploid wheat genome, we have applied several approaches. The RNA-seq data was simulated using gene models of cultivar Chinese Spring using Flux Simulator (http://confluence.sammeth.net/display/SIM/Home). Comparison of transcript abundance estimated for simulated data using Kallisto with actual transcript abundance levels showed a high level of correlation (SCC = 0.98).

We also evaluated the accuracy of Kallisto-based transcript abundance estimates for duplicated homoeologous genes. For this purpose, we simulated RNA-seq datasets using gene models only from one of the wheat genomes (for example, the A genome) and then used all gene models from the wheat reference genome to calculate TPM values. Simulation performed for the A genome showed a high level of correlation (SCC = 0.92, $N = 91,437$) between the real values and those estimated using Kallisto. Only 0.1% of reads simulated using the A genome gene models were mapped to the B and D genomes, indicating high accuracy of transcript abundance estimates for the homoeologous gene sets.

The same RNA-seq simulated dataset was used to estimate the accuracy of read mapping to the correct location in the wheat reference genome using HISAT2[75]. We found that 98% of simulated reads could be unambiguously mapped by HISAT2 to the correct location in the wheat genome.

**SNP genotyping of diverse wheat accessions used for eQTL mapping.** We used a combination of different approaches to obtain genotyping data for the wheat diversity panel: (1) targeted re-sequencing of the regulatory regions of the wheat genome using a Nimblegen capture assay[76], (2) wheat 90 K SNP iSelect assay[77], (3) complexity-reduced genome sequencing[78], and (4) RNA-seq transcriptome dataset. SNPs discovered using the RNA-seq and regulatory sequence capture datasets for 203 wheat accessions were combined, and missing genotype calls were imputed using Beagle[79]. This dataset was combined with the SNPs identified in the entire panel of 400 wheat accessions using the 90 K iSelect assay[77] and complexity-reduced genome sequencing[78]. Further, a common set of SNPs shared between our panel of 400 wheat accessions and 1000 wheat exome dataset[8], were used for genotype imputation (see details below).

Flanking sequences of a genetically mapped set of 46,977 SNPs from the 90 K SNP iSelect assay[77] were aligned to the IWGSC RefSeq v.1.0 using the BLAT program followed by filtering alignments using the following parameters: alignment coverage > 95%, sequence identity >97%, e-value < 1e−10. We identified genomic coordinates for 23,577 uniquely aligned SNPs, which also showed consistency with the marker order in the previously created genetic maps[77]. For these SNP sites, we identified 16,037 SNPs segregating in our wheat panel of 400 wheat accessions.

We have used the wheat regulatory capture assay[76] to re-sequence 203 wheat accessions used for the transcriptome analysis in our study. Up to eight Illumina genomic libraries produced for each sample were pooled together to perform enrichment using the regulatory capture assay. A total of 9,418,016,463 paired-end 2 × 150 bp reads were generated for 203 accessions, with the mean of 46,394,170 reads per accession. Reads were aligned using HISAT2 (v. 2.1.0) with the following parameters: --max-intronlen 70000, --no-spliced-alignment. On average, 87% of all reads were mapped to the genome uniquely, with an average of 8% reads failed to map. The recommended best practices were followed to call SNPs using GATK[80]. Base quality recalibration was performed using genotyping data generated for the same set of lines using the 90 K iSelect assay[77]. The genotype calls for sites with <3 reads depth of coverage were set as missing data. SNPs were filtered to remove sites with more than two alleles, MAF < 0.05, more than 50% genotype calls missing, and more than 3% heterozygote genotypes. In total, we have identified 3,320,006 SNPs segregating in the putative regulatory regions.

For SNP calling, the raw RNA-seq fastq files were processed using the NGSQC Toolkit (v2.3.3) with default parameters. We used HISAT2 (v. 2.1.0) to align reads to the IWGSC RefSeq v.1.0 with the default parameters, except for parameter --max-intronlen set to 70,000. We filtered out reads that are not uniquely mapped to avoid detecting variable sites due to misalignment to the homoeologous genomes. The GATK's 'HaplotypeCaller' was used to generate a *gvcf* file for each sample with the following parameters, '-dontUseSoftClippedBases -stand_call_conf 20.0 '. 'GenotypeGVCFs' was used to generate a multiple-sample VCF file for all variants. Only biallelic sites were used in our analysis. Genotype calls generated for sites with the depth of read coverage less than three or more than 50% genotype missing were set as missing data. Sites with more than 3% heterozygote genotype calls were removed. A total of 2.4 million SNPs were detected in the dataset, of which 138,481 SNPs with MAF > 0.05 were used for analyses.

Construction of complexity-reduced genomic libraries for genotyping the panel of 400 wheat accessions was performed using the complexity reduction protocol, which is based on the digestion of genomic DNA with MseI and PstI restriction nucleases with the follow up ligation of barcoded Illumina sequencing adaptors[78]. The pools of barcoded libraries included up to 96 samples were sequenced on a single lane of HiSeq2500, 1 × 100 bp run. Variant calling was accomplished using Tassel 5 GBS pipeline[81]. A total of 49,150 SNPs with MAF > 0.01 were identified in the panel.

**Genotype imputation.** Genotype data from the 1000 wheat exome project[8] was used as a reference panel for imputation. An integrated VCF file was created, including all samples from 90 K iSelect, complexity-reduced sequencing, RNA-seq, and 1000 exome capture panel. Beagle v. 4.1[79] (beagle.21Jan17.6cc.jar) was then used to impute missing genotype calls with the following settings: 'overlap=500 window=5000 ne=12000'. The genotype calls with probability (GP) <0.8 were considered as missing. Sites with >3% heterozygous genotype calls or >75% missing data were removed, resulting in a set of about 195,000 SNPs.

The VCF files from RNA-seq and regulatory sequence capture datasets were combined into a single VCF file. Imputation was used to fill in missing genotype calls using the same Beagle settings. After imputation, we set genotype calls with GP < 0.8 as missing data. All SNP sites with missing rate >75% or heterozygosity rate >3% were removed, resulting in a set of 4,453,487 SNPs. These SNPs were then merged with the variants identified using the 90 K iSelect array and complexity-reduced sequencing, resulting in a set of 4,449,989 SNPs. A total of 2,021,936 SNPs with MAF > 0.05 in a panel of 198 wheat lines were used for eQTL mapping.

To assess the accuracy of genotype calling, we used genotyping data obtained for our panel using 90 K iSelect array[77]. The genotype concordance rate for different SNP datasets was ~0.98 before imputation and 0.93 after imputation.

For SNP calling using RNA-seq from wheat spikes, fastq files of the previously published 90 RNA-seq samples were downloaded from NCBI (BioProject PRJNA348655: https://www.ncbi.nlm.nih.gov/bioproject/?term=PRJNA348655) using the 'fastq-dump' tool from SRA Toolkit (version 2.9.6). A total of 1.7 million

SNPs were identified using the GATK pipeline. The same settings used for calling variants in the RNA-seq data generated for wheat seedlings were applied to RNA-seq from wheat spikes, except that (1) no imputation was performed, and (2) genotype calls supported by <2 reads were set as missing. After filtering sites with more than 75% missing, 227,922 SNPs with MAF > 0.05 were used for eQTL mapping. The PEER residuals[74] were calculated using the same method used for the seedling RNA-seq dataset. A total of 50,367 HC gene models from the IWGSC RefSeq v.1 were used for eQTL analysis.

**Partitioning genetic variance of gene expression.** To estimate the variance in gene expression explained by different genomes, a set of 2,021,936 SNPs from our panel of 198 wheat lines (GF25, GF32, GF37, GF73, GF270, GF41 lines were removed due to the low proportion of mapped RNA-seq reads to the reference genome; Supplementary Data 1) and 227,922 SNPs from a set of 90 lines were grouped into three genome-specific sets (A, B, D genomes). Each set was used to build genetic relationship matrices using '--autosome-num 30 --make-grm-inbred' in GCTA[25]. The genetic variance of expression traits was then calculated for three subsets jointly using '--mgrm --reml'. Out of the top 10,000 genes showing the highest levels of expression variance, 8698 gene expression traits in seedlings and 7090 gene expression traits in spikes were successfully processed (log-likelihood converged.) To remove the confounding effect of SNP density in different genomes, we used 1 SNP per 100 kb genomic window for the calculation of the genetic relationship matrix.

**Detection of eQTL.** The association between SNPs and gene expression PEER residuals was performed by Matrix eQTL (v. 2.1.0)[82] with the setting 'useModel = modelLINEAR'. The set of 191 accessions having matching RNA-seq and SNP genotyping data was used for final eQTL mapping (Supplementary Data 1). In addition to 6 lines removed due to low RNA-seq mapping quality (see above), we also removed 7 lines (GF294, GF342, GF366, GF380, GF381, GF383, GF387) that showed lack of good genotyping data concordance in the panel. The first three principal components (PCs) of the SNP matrix were used as covariates. Based on the estimates of genomic inflation factor (GIF), this approach was effective in controlling population structure for nearly 61% of genes, which showed no evidence of inflation of test statistics (61% of genes had GIF < 1.1) (Supplementary Fig. 16). While the remaining genes showed some effect of population structure on test statistic, these effects did not substantially inflate false discovery rate assessed by permutation of phenotypic data relative to genotypes. The expression values of each of the 52,060 genes in our seedling dataset were permuted relative to genotyping data (includes 2,021,937 SNPs) to generate 1000 randomized datasets. The SNP-gene expression association test statistic was calculated using Matrix eQTL. By applying p-value threshold corresponding to FDR ≤10−5, on average, we detected 3,595 associations in the randomized datasets. In the real-life dataset, we have identified 11,421,859 associations (before LD merging) passing this significance threshold indicating that only $3.2 \times 10^{-4}$ associations passing our threshold are false positives. A similar permutation approach was applied for assessing the proportion of false positives among detected eQTL in the spikes. While in the original non-permuted dataset, 1,336,626 SNPs pass this p-value threshold (before LD merging), in the permuted datasets, on average we had only 10,858 SNPs passing threshold, suggesting that in spike eQTL the actual false discovery rate is around $0.8 \times 10^{-3}$.

All associations with FDR < 1e−5 were considered as significant. For each transcript, significantly associated SNPs were merged based on LD ($r^2 > 0.2$) and distance (<100 kbp) into genomic intervals. SNP with the strongest association signal within an interval was defined as an eQTL of the transcript. If an eQTL was located within ±1 Mb around the target gene, it was defined as *cis*-eQTL. In our analyses we used a conservative definition of *trans*-eQTL, which was an eQTL significantly associated with the target gene located on a different chromosome. The eQTL effect size estimated by Matrix eQTL is based on the linear regression slope.

**Analysis of eQTL and Hi-C data.** Hi-C data for the hexaploid bread wheat cultivar Chinese Spring was downloaded from the NCBI database[28]. The Juicer Tools (v1.21.01) was used to process all valid read pairs downloaded from NCBI GEO (GSM3929163_Wheat.shoot.hicpro.allValidPairs.txt.gz). We first generated a.hic file using the 'pre' command, then the observed contact frequency map was calculated at 1 Mbp resolution using the 'dump' command without normalization. The 3D chromatin contacts between a pair of eQTL-eGene was estimated based on the contact frequency between the two 1 Mbp genomic intervals. For example, if eQTL is located at position chr7A_20102690 and its eGene TraesC-S1A01G002200.1 is located at a position chr1A_1188779, their Hi-C contact frequency was estimated between the genomic intervals 20–21 Mbp on chromosome 7A and 1–2 Mbp on chromosome 1A. We did not analyze the 3D chromatin contact between *cis*-eQTL and its eGene directly due to the relatively low depth of read coverage in the downloaded data. In order to evaluate the significance of the observed Hi-C contacts between *trans*-eQTL and eGene, we generated randomized distribution of Hi-C contact frequency between randomly selected pairs of genomic intervals.

**Phenotyping wheat for productivity traits**. We have used field-based phenotyping data previously collected for a diverse panel of about 800 wheat accessions from the 1000 wheat exomes project[8], including grain filling period (GFP), harvest weight (HW), drought susceptibility index for harvest weight (HWS), heading date (HD) and plant height (PHT) traits. Field data for these accessions were collected for two consecutive years under rainfed and irrigated conditions at the Agriculture Victoria research station located at Horsham, Victoria, Australia. Three replications of each accession were planted in 4.5 m single rows in a randomized block design, with a seed spacing of 3.6 cm and row spacing of 65 cm. HD was recorded as the date on which 50% of the heads in a row fully emerged from the culms. PH was measured from the ground to the tip of the spike, excluding awns. The Best Linear Unbiased Estimates (BLUEs) were obtained using a model with fixed genotype effects and all other effects set as random in an individual year. The trait values from the rainfed and irrigated fields were used to calculate the drought susceptibility index for harvest weight (HWS), according to Fischer & Maurer[83].

A panel of 400 spring wheat accessions were selected to represent genetic and geographic diversity of wheat (Supplementary Data 1). For phenotyping, plants were grown in Kansas State University greenhouse with 16 h light/8 h dark conditions with temperature set to 21 °C during the night and 24 °C during the day. Three plants of each accession were grown in the 1 gallon round pots filled with a self-made soil mix (volume ratio was 20 soil: 20 peat moss: 10 perlites: 1 CaSO$_4$). Plants were arranged according to a complete randomized design. Phenotyping data collected for the panel of wheat lines is listed in Supplementary Data 8. The heading date (HD) data was collected for two planting seasons when the plants reach stage 50, according to Zadoks scale[84]. The date when the first spike in a pot appeared from the flag leaf sheath was recorded. The awn length (AWN) was measured after ripening. The accessions without awn, with short awn, and long awn were given scores 0, 1, and 2, respectively. The data were collected for four planting seasons. The plant height (HT) was measured after ripening, from the base to the top of the main stem. The height of three plants for each accession was measured each season for four planting seasons. The spike length (SPL), spikelet number per spike (SPN), and spike compactness (SPC) measurements were collected from the main spike of three plants for each accession. The data was collected for three planting seasons. Grains from all the spikes of each plot were harvested and used for data collection. The MARVIN seed analyzer (GTA Sensorik GmbH, Germany) was used to estimate the Thousand Grain Weight (TGW) and grain width (GW), length (GL), and area (GAR). In addition, the grain length to width ratio (GWLR) was calculated by dividing the grain length by grain width. The grain morphometric phenotypes were collected for three planting seasons.

**Summary data-based Mendelian randomization analysis**. We used GCTA (v. 1.92.2beta) to perform genome association mapping[85] in a panel of 400 wheat lines. The genetic relationship matrix was calculated using the command '--make-grm-inbred --autosome-num 30'. The best linear unbiased predictions (BLUPs) were calculated for each phenotype by fitting a mixed linear model using the *lmer* function of R[86], and used for association mapping. On average, SCC between BLUPs and mean phenotypic values was 0.99. The '-mlma' was used to calculate the association between SNPs and phenotypes. We used the first three principal components to control for population structure.

We applied summary data-based Mendelian randomization analysis (SMR)[29] to evaluate the association between gene expression and trait variation using summary-level data from our eQTL mapping study and two GWAS, one of which was performed in the current study, and another one was accomplished within the 1000 wheat exome project[8]. The summary-level statistic of these two GWAS datasets was analyzed using the SMR commands '--trans-wind 5000 --diff-freq-prop 0.2' for *trans*-eQTL and '--diff-freq-prop 0.2 --cis-wind 10000' for *cis*-eQTL. The eQTL identified using RNA-seq data from wheat seedlings and spikes were analyzed separately. Heidi test was used to separate functional associations from association due to linkage[29].

**Construction of GCN connected with the SMR gene candidates**. The candidate genes for each trait from SMR analysis (SMR *p*-value < 1e−4) based on seedling stage eQTL were used as a starting list of genes for network construction. In total, we used 1899 SMR gene candidates identified using the eQTL data identified in the wheat seedlings and summary-level data from GWAS conducted in the wheat 1000 exome project[8]. We obtained co-expressed genes (|SCC| > 0.6) among the 198 seedling RNA-seq samples for all SMR gene candidates. All genes co-expressed with |SCC| > 0.6 with the SMR gene candidates or genes connected with the SMR gene candidates were included in this network. SMR candidate genes showing no co-expression with other genes in our dataset were excluded from this co-expression network. The Gephi (https://gephi.org/) was used to visualize the GCN, which contains 3642 nodes and 57,837 edges.

In addition, we used the seedling stage RNA-seq dataset to infer regulatory relationships between *cis*-eGene and its associated *trans*-eGene that are showing high levels of expression correlation at |SCC| > 0.6. For this purpose, we used all SNP-expression associations (FDR < 1e−5), which included 21,354,094 associations between 1,840,991 SNPs and 32,679 gene transcripts. We assumed that a SNP associated with both a *cis*-eGene and a *trans*-eGene at the same time, there is a potential regulatory relationship that exists between these two genes (Fig. 5a), and

*trans*-eQTL effects are observed due to variation in the expression of a *cis*-eGene that acts as trans-factor. This scenario could be applied to the cases where a *cis*-eGene is a transcription factor, and a *trans*-eGene is its regulatory target. In total, we detected 19,186 pairs of putative regulatory interactions among 5150 genes. The SCC values calculated using the PEER residues for each pair of genes showed a bimodal distribution (Supplementary Fig. 9). On the contrary, the background distribution of SCC values for random pairs of genes was unimodal, with a peak centered around 0. If the absolute value of SCC was larger than 0.6, we predicted a potential regulatory relationship. In total, we predicted 1,200 regulations between 536 regulators and 294 targets. There were 903 negative regulations (SCC < −0.6) and 297 positive regulations (SCC > 0.6). Densely connected clusters of genes (network modules) were identified using a default routine implemented in Gephi[87]. The network modules were also tested for GO enrichment (Supplementary Fig. 10, Supplementary Data 13–15). The network was also supplemented by information about genes previously characterized in wheat or rice[65].

**Correlation between productivity traits and number of low-expressing alleles from the homoeologs with the biased expression**. Among 21,807 homoeologous genes used in our eQTL analysis, we have identified 59 homoeologs showing evidence of biased expression dosage. We defined biased homoeologous genes (or negatively correlated homoeologs) using the following criteria: (1) negative correlation (SCC < 0) with both of its two homoeologs from other genomes, and (2) strong negative correlation (SCC < −0.4) with at least one of its homoeologs. In most cases, negative inter-homoeolog expression correlation among these 59 homoeologs was associated with the presence of accessions in the population that carry the downregulated gene variants. In each accession, a homoeologous gene was considered downregulated if it showed TPM < 3 and TPM < mean - stdev, where 'mean' is the average TPM value among all wheat lines, and the 'stdev' is the standard deviation of TPM values among all wheat lines. Correlation between productivity trait BLUPs and the total number of low-expressing alleles from this set of 59 homoeologous genes in each accession was estimated using SCC. These SCC values were compared with the distributions of SCC values calculated between 1000 random sets of 59 homoeologous genes and each of the productivity traits.

**Model-based prediction using the expression matrix of 59 homoeologous genes showing biased expression**. Model-based prediction of productivity traits was performed using ridge regression method implemented in R package 'glmnet'[88] using gene expression matrix with TPM values of 59 negatively correlated homoeologous genes for 198 wheat lines. Tenfold nested cross-validation was performed to test the accuracy of predictions[35]. To assess the association between negatively correlated homologous genes and productivity traits, the accuracy of predictions obtained using these 59 genes was compared with the prediction accuracy obtained using the random sets of 59 genes from homoeologous gene triplets.

**Reporting summary**. Further information on research design is available in the Nature Research Reporting Summary linked to this article.

## Data availability
RNA-seq and genome resequencing data generated in this study are deposited to NCBI SRA PRJNA670223, PRJNA787276 and NCBI GEO GSE167479. All analyses were conducted using standard software. The settings of software used for analyses are described in the Methods. Source data are provided with this paper.

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

## Acknowledgements

This research was supported by Bill and Melinda Gates Foundation (INV-004430) to M.R., E.A., A.A., L.S., by the Agriculture and Food Research Initiative Competitive Grants 2022-68013-36439 (WheatCAP) to J.D. and 2019-67013-29017 from the USDA National Institute of Food and Agriculture to E.A., and by the International Wheat Yield Partnership (IWYP). We thank Dr. Li Lei for comments on an earlier version of the manuscript.

## Author contributions

F.H.—bioinformatical and statistical analyses of data; F.H., W.W., J.D., P.L.M., and E.A.—data interpretation; M. Rouse— assembled diversity panel; W.W., M. Rouse, L.S., E.T., N.D., D.S., S.S., S.D., M. Reynolds, J.H., S.K.S., S.L., J. Chen, A.F., J. Cook, G.B., A.F., M.P., A.X.C., M.S., L.S., and M.J.H.—phenotypic characterization of the wheat populations and trait mapping, W.W., K.W.J., and A.A.— complexity-reduced sequencing and analysis; W.W., K.W.J., and M. Rouse—array-based SNP genotyping and analysis; W.R.—RNA-seq library construction and initial data processing; A.A. and J.R.—targeted sequence capture and RNA-seq data generation; F.H., W.W., P.L.M., and J.D.—editing the first draft of the manuscript; E.A.—conceived idea, performed data analyses, coordinated data collection, and wrote the manuscript.

## Competing interests

The authors declare no competing interests.
