## [Peer Review File · Nature Communications]

Genomic variants affecting homoeologous gene expression dosage contribute to agronomic trait variation in allopolyploid wheatReviewers' Comments:

Reviewer #1:

Remarks to the Author:

In this manuscript, He and colleagues used diverse allohexaploid wheat accessions to map expression quantitative trait loci (eQTL) and evaluated their effects on the population-scale variation in homoeolog expression levels. Authors used association mapping to identify cis- and trans-acting variants that explain the variance in homoeologous gene expression, with gene expression treated as a phenotype. To study the effect of chromatin dynamics on regulatory variants, authors investigated the distribution of cis- and trans-acting eQTL across genomic regions showing distinct chromatin architecture or involved in 3D chromatin contacts. Finally, eQTL, GWAS and gene co-expression networks were analyzed jointly to investigate the role of variants linked with homoeologous gene regulation in shaping variation in major agronomic traits in wheat landraces and cultivars. Overall, the topic is interesting but the work is only descriptive and the part integrating chromatin dynamics and chromatin architecture is not convincing.

Major Points:

1. Authors showed that cis- and trans-eQTL were found enriched in the regulatory regions however it is not shown if any mutation can impact chromatin accessibility. Performing some ATAC-seq on contrasted wheat varieties to connect some genetic variation with variation in chromatin accessibility would really improve the manuscript.
2. The analysis correlating histone modifications (H3K4me3, H3K4me1, H3K27ac, H3K27me3 and H3K9me2) and cis- and trans-eQTL does not add anything to the paper since no one of those histone marks is clearly associated to regulatory region. All of those marks are mainly associated to gene ORF or TE for H3K9me2.
3. Authors showed that both cis- and trans-eQTL p-values positively correlate with the frequency of Hi-C contacts. A major point here is the resolution of the Hi-C data used that is not sufficient for a precise mapping of the interaction. In addition none of the interaction was validated by another method like FISH or 3C experiment. In addition to prove that some eQTL effects likely depend on the physical interaction between the regulatory elements and target genes, authors should at least demonstrate that some loops are destabilized in genetically diverse allohexaploid wheat.
4. Authors raised the possibility that the effects of trans-eQTL on target genes located in the homoeologous regions could be mediated by chromatin loops but again here this is supported by only the correlation between trans-eQTL and both homoeologous and non-homoeologous chromosomes. To my view this is not enough (see point 3)

Reviewer #2:

Remarks to the Author:

In the manuscript entitled "The landscape of genetic effects on gene expression levels in allopolyploid wheat reveals the impact of homoeologous gene dysregulation on agronomic traits," He et al. investigated the genetic control of gene expression (a.k.a. expression quantitative trait loci or QTLs) in allopolyploid wheat. They observed uneven distribution of genetic variations for gene expression in different homoeologous genomes, likely owing to polyploidization and selection. They showed an intriguing inverse relationship between genetic variance for the expression level of homoeologous genes and expression correlation between the homoeologous gene pairs. They also showed that homoeologous gene regions are depleted for both common and rare genetic variants but enriched for Hi-C contacts. They further demonstrated the integration of the eQTL data with GWAS data to discover functional genes for agronomic traits. The design of the experiment and the most methods used are sound. The manuscript is well written. However, I have a few major concerns regarding the method used for eQTL analysis and the choice of statistical significance level for claiming eQTLs.

Major comments

The authors used a linear model method implemented in the Matrix eQTL software for eQTL detection.

First of all, and most importantly, it is unclear how the cryptic relatedness in the sample was accounted for and the extent to which the eQTL test statistics were inflated due to the cryptic relatedness. It is also unclear whether three eigenvectors are sufficient to control for population stratification. The authors may need to demonstrate the distribution of the genomic inflation factors for all the gene expression traits analyzed. One suggestion would be to use a linear mixed model method that can account for both relatedness and population stratification, as the authors did in GWAS analysis for the agronomic traits.

Another critical issue is the choice of statistical significance level for claiming eQTLs. The authors chose an FDR threshold of $1e-5$ without justification. However, the number of false positives in the reported eQTLs is unclear, especially considering that relatedness has not been accounted for in the eQTL analysis.

This is reflected by the large number of eQTLs detected (after LD-based merging) from both the seedling and spike data sets, irrespective of the small sample sizes. Although the sample size of the seedling data is more than twice that of the spike data, the number of eQTLs detected from the former (36,898) is substantially smaller than the latter (65,117). Interestingly, after integrating the eQTL data with the GWAS, the number of trait-associated genes using the seedling eQTL data (329) was much larger than that using the spike eQTL data (95), consistent with the difference in sample size between the two data sets.

Several observations might indicate that there are a substantial number of false-positive trans-eQTLs. 1) One observation is related to the expression correlation. If genetic variations are expected to disrupt the expression correlation between homoeologous genes, then the expression correlation between homoeologous gene pairs that share eQTLs is expected to be smaller than that for random gene pairs (Figure 4). This is the case for genes that share cis-eQTLs but not for those that share trans-eQTLs. On the other hand, the bimodal distribution (e.g., the mode on the right-hand side) may indicate the enrichment of false positives. 2) The second observation is related to overlap between Hi-C contacts and trans-eQTL-target gene pairs. As presented in Figure 5d, the density plot for trans-eQTL-target gene pairs is highly similar to that for a random set of genes. BTW, I do not think the test has been done correctly in this case. A proper test should be a test of the mean (mode or median) of the number of Hi-C contacts for the trans-eQTL-target gene pair against the distribution of the mean (mode or median) values obtained from repeated sampling of random gene sets (rather than one random gene set). 3) The third observation is the lack of overlap between the trans-eQTLs and the QTLs for the agronomic traits. 4) The proportion of genes with trans-eQTLs that also have cis-eQTLs is surprisingly low, especially in the seedling data (1469 / 8315). Is it also an indication of the potentially elevated false-positive rate?

Ln320-321. The observed negative correlation could be due to ascertainment. The statistical power to detect an eQTL is a function of $n * 2p(1-p) * b^2$ with n being the sample size, f being the allele frequency and b being the effect size so that the effect size of a lower MAF variant needs to be larger to be detected at a specific significance level.

Ln855-857 and Ln887-889. The authors may need to clarify the purpose of using BLUEs and BLUPs of the phenotypes for follow-up analyses and why they used BLUE for the 800 accessions and BLUP for the 400 accessions.

Ln956 "Ten-fold cross-validation". This method is only applicable to data where all the individuals are independent.

Minor comments

Ln197 "Only half of these genes". Please be specific about "these genes".

Figure 2c. The authors may need to clarify how this plot was made because genetic variance is

estimated for a single gene, whereas expression correlation is computed for a pair of genes.

Ln560 Fig. 7f?

Reviewer #3:

Remarks to the Author:

The authors investigated the regulatory control of duplicated gene expression in hexaploid bread wheat from a popgen perspective and its relevance to agronomic traits:

1. partitioning genetic variation of expression traits to evaluate regulatory control from the same (cis-acting variants) or different (trans-acting) subgenomes, revealing stronger effects from former. Line 244 concluded that "dysregulation of homoeologs is primarily associated with the cis-regulatory diversity": what does "cis-regulatory diversity" refer to and how is it connected with the cis-acting variants?
2. eQTL analysis identified cis-eQTLs and trans-eQTLs, which were next annotated based on chromatin features: trans-eQTLs were further examined to explore the possibilities that homoeologous genes act upon each other in trans; cis-eQTLs were compared between homoeologs and singletons to ask whether duplicated genes are under more relaxed purifying selection; both were next considered together in four configurations to examine their effects on relative levels of homoeolog expression.
3. Agronomic traits analysis: this is where I had to stop due to the lack of clarity of the above two aspects, detailed below.

So there are A LOT of good ideas and tests involved, and I do believe the authors are onto some quite novel aspects on the regulatory complexity in allopolyploid plants, but also because of such complexity, this manuscript is not easy to follow. The title is rather lengthy but doesn't have a clear point. The abstract appears to cover main results, but these results are disconnected to deliver a cohesive story.

One major hurdle for me to understand this manuscript is the concept of "homoeolog dysregulation". According to Figure 1a, the authors used this term to describe homoeologs that are differentially expressed, which doesn't indicate any functional consequences like impairment in protein functions or metabolic process. In that case, there are already terms like "unequal expression" and "homoeolog expression bias" for that. But later, "dysregulation" was defined more specifically based on negative SCC. All these terminological and conceptual inconsistency brought by the new term make it difficult to connect this work with relevant literatures to comprehend new evidence and findings.

Another issue is the use of cis and trans, more detailed comments are marked in the attached PDF.

Methods seemed sound to me and carefully executed for each task. For example, I do appreciate the efforts the authors put in to assess transcript estimation, which employed simulated RNA-seq data to validate the high correlation between observed and expected reads.

Reviewer #4:

Remarks to the Author:

Overview:

The authors investigate the causes and consequences of differential expression of homoeologs among lines from a wheat diversity panel and attempt to associate expression changes with genotype and trait variation. They make a number of findings. They identify balanced homoeologs as those that have the same subgenome contribution in all wheat lines and dysregulated ones as those with where subgenome contributions differ by line. For many homoeologs, expression is balanced. The authors

find incomplete dosage compensation for dysregulated homoeologs. For genes on the A and B subgenomes, expression variance is largely attributed to the native subgenome whereas for genes on D, which arose after a bottleneck event, all three subgenomes contribute equally for expression variance. Balanced homoeologs are associated with fewer genetic differences among lines and share more eQTLs as expected for more consistent expression. Cis and trans eQTLs associated with expression variation have been attributed to functional and regulatory regions. Homoeologous gene regions also seem to have more contact with each other. Among many other findings, they also find that accumulation of low expressing alleles in dysregulated homoeologs is associated with trait differences between wheat lines.

Previous work has identified cis and trans eQTLs polyploid species including polyploid Arabidopsis (Shi et al. 2012), potato (Zhang et al. 2020) and cotton (Bao et al. 2019) but this study would be the first to identify them in wheat and study how they interact to affect homoeolog expression. The study also attempts to associate dysregulated expression of homoeologs with trait values. However, this dysregulation-trait association is less robust compared to the genetic partitioning and eQTL analyses.

Bao Y, Hu G, Grover CE, Conover J, Yuan D, Wendel JF. Unraveling cis and trans regulatory evolution during cotton domestication. *Nature communications*. 2019 Nov 27;10(1):1-2.

Zhang L, Yu Y, Shi T, Kou M, Sun J, Xu T, Li Q, Wu S, Cao Q, Hou W, Li Z. Genome-wide analysis of expression quantitative trait loci (eQTLs) reveals the regulatory architecture of gene expression variation in the storage roots of sweet potato. *Horticulture Research*. 2020 Jun 1;7(1):1-2.

Shi X, Ng DW, Zhang C, Comai L, Ye W, Chen ZJ. Cis-and trans-regulatory divergence between progenitor species determines gene-expression novelty in Arabidopsis allopolyploids. *Nature communications*. 2012 Jul 17;3(1):1-9.

Major Comments

Is the term "balanced" accurate for what is described in terms of homoeolog expression? Balanced in previous work (e.g. Ramirez-Gonzalez et al., 2018, Science) refers to similar expression levels of A, B and D homoeologs e.g. all three are expressed at 1 TPM. My understanding is that here by "balanced" the authors mean that the expression level of the homoeologs is correlated but not necessarily at the same expression level e.g. if the ratio between A:B:D is 4 TPM: 1 TPM: 1 TPM and that is consistent across the population (SCC will be positive) the relationship is considered "balanced". This seems non-intuitive based on the terminology. Would "positively correlated homoeologs" be a more suitable term to avoid confusion with previous work in this area? The graphic in Fig 1A should be updated using multiple possible relationships: currently the nuance about the correlated expression is lost when relying on that figure for definitions. (i.e. include a line at 1:1 and a line at e.g. 3:1 to show the expression level of A:B does not have to be equal).

The term "dysregulated" might be more accurately described as "negatively correlated homoeologs". This is what is described in the methods (line 393-942) which says a dysregulated homoeolog should have a negative correlation ($SCC < 0$) with two homoeologs, and a strong negative correlation ($SCC < -0.4$) with at least 1 homoeolog. However, the Figure 1 legend describes "dysregulated homoeologs show different levels of expression" which to me means something quite different (i.e. A expression is higher than B expression). The explanation of dysregulated homoeologs should be made clearer and consistent throughout the paper.

Similarly, the definition and consistency of the 59 homoeologs identified as being dysregulated could be made clearer. Was the level of dysregulation and homoeolog expression exactly the same between

seedlings and spikes for each line? More generally, how consistent are dysregulated homoeologs across tissue, time and biological replicate? It not evident that expression at this timepoint being studied is the only one affecting trait expression. If homoeolog contribution changes with development, it is possible other homoeologs are identified as dysregulated at a timepoint or tissue not investigated here and may have a more direct impact on trait values. Ideally, the tissues underlying the basis for a given trait should be studied (e.g. grain gene expression for a grain related trait) but this study provides a first pass attempt at trying to identify associations. This limitation should be discussed, and support for the approach used comes from work in maize which shows that several different tissues can be used to predict seed-weight (Kremling et al., 2018).

Kremling, K., Chen, SY., Su, MH. et al. Dysregulation of expression correlates with rare-allele burden and fitness loss in maize. *Nature*. 2018. 555, 520–523

The conclusion (line 243-245) that “our results suggest that the dysregulation of homoeologs (Fig. 1a) is primarily associated with the cis-regulatory diversity” is not fully supported by the evidence presented. The majority of the results do support that cis-genomic diversity is associated with homoeolog dysregulation, but this does not go as far as supporting “cis-regulatory diversity” because the analysis was done at a subgenome scale (e.g. A vs B vs D), rather than at a specific chromosome scale. Therefore, I find the current statement misleading because cis-regulatory would normally refer to a region close to the gene being regulated (or at the least on the same chromosome) but in this case most of the data shows the regulation comes from the same subgenome, but not necessarily the same chromosome. A way to improve this analysis, could be to re-run the partitioning of genetic variation per chromosome, rather than per subgenome, then the data could support this conclusion. Otherwise, the conclusion should be amended to “our results suggest that the dysregulation of homoeologs (Fig. 1a) is primarily associated with the cis-genomic diversity”.

Three comments related to the assessment of purifying selection homoeologs and singletons in Fig 4d:

a) There is no explicit test for purifying selection in Fig 4d. One way to do this, as done for many of the other analyses in this paper, is to permute the expression and genotype, separately for each MAF category, and establish the significant threshold.

b) Related to the same analyses, the effect size for eQTLs with low MAF can be inflated for a number of reasons and detecting rare small effect QTLs in general is difficult. It would be worth repeating the analyses using a fixed contribution of two alleles by subsampling genotypes for the more common allele and repeating the effect size calculations. The results of Fig S2C suggests this is unlikely to alter conclusions regarding the comparison between singletons and homoeologs.

c) Why not consider doublets and triplets separately?

Methodological major comments:

It would be important to know if the key conclusions still hold when excluding genes that show high levels of expression noise – variability between biological replicates. Since the three samples together were ground together, this current data does not allow one to assess this.

Is Hi-C stable between lines? Hi-C data from CS is being used to infer relationships in all lines, yet they are very different genotypes which are likely to have large structural re-arrangements. Therefore, whilst interesting, these results are a bit speculative. How robust are comparisons when Hi-C datasets from different genotypes are used? One way to test this would be use the datasets generated Walkowiak et al. (2020) and test if the results hold under all conditions.

Walkowiak S, Gao L, Monat C, Haberer G, Kassa MT, Brinton J, Ramirez-Gonzalez RH, Kolodziej MC, Delorean E, Thambugala D, Klymiuk V. Multiple wheat genomes reveal global variation in modern

breeding. Nature. 2020 Nov 25:1-7.

Line 117-119 "Transcripts Per Million (TPM) were estimated for high-confidence (HC) genes in RefSeq v.1.0, with 82,092 genes (66,333 genes) showing TPM > 0.5 in at least two wheat lines". But this does not agree with what is stated in the methods (Lines 701-704) which states that "Gene models with expression standard deviation > 0.5 and expressed (TPM > 0.5) in at least three samples have been used in our analyses. This set of genes included 52,511 HC gene models, 29,226 LC gene models, and 13,861 de novo assembled transcripts." Were only HC genes used, or were LC and de novo transcripts also included?

Line 700. How were the de novo transcripts assembled? And which analyses were they used for?

Data availability:

Deposited RNA-seq data is private until 1st Nov 2023, this must be made public upon publication. Also, this is not the raw RNA-seq reads (is this in the linked SRA record? I cannot access it to check). The raw fastq files should also be made available upon publication. Finally, these SRA and GEO accession numbers should be included in the main methods or in list of supplemental data so that others can find them.

Minor comments:

Title is very long. Could it be re-written to be shorter and more informative?

Are the 2Mb regions defined surrounding gene always intergenic regions? If closer regions (500 bp and 1 MB) are less likely to intersect with neighbouring genic regions, they could be used instead.

Text on figure 1 is too small to read. Figure 1B does not really add to the narrative and could be removed.

In Fig 3A and B, what does deleterious mean? Missense can also be deleterious.

Figure 5, text too small panel e). Explanation of tpm graphs on the right not clear. Also why is this panel inside a blue box?

Figure 7C, the Y axis is misleading, it's a count of low expression alleles for dysregulated homoeologs in each line, or so I think.

Is Line 560 supposed to say Fig 7F?

Line 593 "Previous studies in maize" – only 1 study is cited.

Line 683. Which programme and parameters were used for read mapping? Which genome sequence was used as a reference?

Line 784 "All SNP sites with missing rate > 75% or heterozygosity rate < 3% were removed". Why remove SNPs with low heterozygosity? I would expect most wheat lines to be homozygous. (This is also the opposite of the stated filter on line 779)

Line 926 to 928 there is a sentence repeated "If the absolute...."

Supplementary table 2, the column named "GENE_ID (v.2.0)" should be called "Gene_ID (v1.1)", consistent with IWGSC et al., 2018.

Supplementary table 12 is labelled as Supplementary Table 13 in the excel file so needs to be corrected.

Supplemental Fig 14. What are eGenes?

Response to the Reviewers' Comments

Reviewer #1 (Remarks to the Author):

Remark from the authors: We would like to thank this reviewer for the critical comments. Although, nearly all of them are targeting only small secondary direction of analyses concerning chromatin, which was never intended to be something of central importance for our manuscript, these comments forced us to more critically re-assess the analyses we conducted to study the relationship between eQTL and chromatin. Based on this assessment, we have removed some of the Hi-C analyses from the manuscript and also restructured and revised our manuscript trying to present more careful interpretation of obtained results.

Major Points:

Comment 1: Authors showed that cis- and trans-eQTL were found enriched in the regulatory regions however it is not shown if any mutation can impact chromatin accessibility. Performing some ATAC-seq on contrasted wheat varieties to connect some genetic variation with variation in chromatin accessibility would really improve the manuscript.

Response: While we do agree that detecting eQTL that control gene expression by changing the chromatin states would be an interesting study, we do not believe that adding this new quite large datasets and associated results (please see below why mapping SNPs affecting chromatin states would require much larger scale research than just analyzing few contrasting lines with ATAC-seq) would improve the main messages in our current manuscript. Though we have added a new analysis of the previously published ATAC-seq and RNA-seq data that points at possible connection among eQTL, chromatin states and gene expression (you may see Results and Methods section of the revised manuscript), we did it reluctantly. Below we provide our opinion about the new analyses proposed by the reviewer.

- 1) Detection of eQTL variants that specifically act through mechanistic modification of chromatin to regulate expression is quite complicated research task that is surely outside of the scope of our current manuscript. In our manuscript, for the first time, we characterized a) eQTL and their effects on homeologous gene expression in polyploid species, b) the role of the polyploid genome's evolutionary history on the cis-/trans-effects across homeologous chromosomes, c) the relative role of cis-/trans-eQTL in homeolog expression bias, and d) the impact of homeolog expression dosage variation on agronomic traits targeted by breeding. We are concerned that addition of these new quite substantial analyses of eQTL/chromatin states/gene expression would not leave enough room to adequately describe these main findings, most of which reported for the first time in a polyploid species. The other three reviewers found that our manuscript is already too complex and would benefit from simplification and shortening. We do not see how we could accomplish this by further expanding manuscript and supplementing it with additional data, which we would not be able to describe in sufficient details either.

- 2) We also believe that this is misleading statement that it is easy to identify eQTL that act on genes by changing chromatin states by simple comparison of some contrasted wheat varieties. Essentially, this reviewer asks us to identify causal variants that are both eQTL and caQTL (chromatin accessibility QTL). While selection of several contrasting accessions and running ATAC-seq can help to find an eQTL allele located near genes showing different chromatin states, this design will not be able to prove that changes in chromatin states are due to this particular eQTL. Due to small sample size in this experimental design, we would not be able to rule out that chromatin accessibility is controlled by other variants located elsewhere on a chromosome and in LD with the chromatin state around the eQTL region. To separate the effects of eQTL from other variants on chromatin states, one would need to analyze much larger population, where LD range is much shorter than that captured within few wheat accession. Thus, mapping caQTL by ATAC-seq would require a panel of lines comparable in size to our eQTL mapping panel, at least 100 or more lines. Putting aside that this is not the main focus of our work, due to the size of the wheat genome (16 Gb) and simply the large cost of experiment and time needed to perform it, we do not feel that the characterization of caQTL-eQTL pairs should be the focus of this first eQTL mapping study in wheat. In addition, due to COVID-related issues that forced us and our genomics and cell-sorting facilities to limit in-person work in the laboratories, conducting additional genomics experiments such as ATAC-seq is nearly unfeasible.
- 3) The enrichment analyses in genomics research are powerful tool and have specific purpose within the scope of large-scale variant mapping studies, such as eQTL mapping. The purpose of enrichment analysis per se is to statistically detect the existence of relationship between variables and, in eQTL mapping setting, also to serve as a quality control to confirm that methods applied for eQTL mapping could detect variants located in the regions of genome potentially involved gene expression regulation, which includes promoter regions upstream of genes that also tend to have open chromatin. As such, they do not normally include demonstrating that any given specific mutation changes chromatin state or not, which is the goal of more focused functional studies pursuing this specific objective. The fact that eQTL are over-represented in the regions of open chromatin compared to randomized set is sufficient to demonstrate that detected eQTL set, in general, is preferentially located in functionally active regulatory regions of genome. For example, the human GTEx consortium (Aguet et al. Genetic effects on gene expression across human tissues. *Nature*. 2017;550:204–13) used overlap with DHS regions to functionally annotate their detected eQTL. Thus, for one of the first eQTL mapping studies in wheat this specific enrichment analyses should have been performed. But more in depth analysis of the relationship between eQTL, chromatin state and gene expression is the topic of follow-up studies. Our study is already quite extensive in scope of reported analyses and, after obtaining comments from other reviewers who asked for shortening and simplifying the manuscript, we feel that we would do poor service to the research community by further expanding analyses to yet another direction.
- 4) Though we believe that this does not substantially strengthen main conclusions in the manuscript, to partially satisfy this reviewer's request, we tried to expand eQTL-

chromatin state analyses for a selected subsets of eQTL from our study into two cultivars for which we have published ATAC-seq data available: cv. Chinese Spring and Paragon, both were not part of the analyzed panel of lines. Our results indicate that about 17% of genes showing differential gene expression between these two cultivars with the eQTL allele effect directions consistent with the direction of expression change also show expected changes in chromatin accessibility of the promoter regions. The Results and Methods section are updated, and data is shown in Supplementary Table 6.

Comment 2. The analysis correlating histone modifications (H3K4me3, H3K4me1, H3K27ac, H3K27me3 and H3K9me2) and cis- and trans-eQTL does not add anything to the paper since no one of those histone marks is clearly associated to regulatory region. All of those marks are mainly associated to gene ORF or TE for H3K9me2.

Response: It is not clear to us why this reviewer states that H3K4me3, H3K4me1, H3K27ac and H3K27me3 are not associated with the regulatory regions? In the original version of the manuscript, we did provide in the Results section brief description of the generally accepted role of each mark in regulation. The H3K4me3, H3K4me1, H3K27ac marks are well known for being associated with enhancers and promoters of actively expressed genes. The H3K27me3 marks are often linked with repressed chromatin and are enriched in the promoter regions of repressed genes. Only one of the epigenetic marks listed by the reviewer, H3K9me2, is not associated with the regulatory regions (and we did indicate this in the Results section too). Here we provide additional short citations (now added to the manuscript) from published studies that specifically mention association between H3K4me3, H3K4me1, H3K27ac and H3K27me3 and the regulatory regions: 1) “H3K4me1 is highly enriched at enhancers, ... and H3K4me3 is a hallmark of the promoters of actively transcribing and poised genes” (DOI: <https://doi.org/10.1038/emm.2017.11>); 2) “Selected epigenomes also contain a subset of additional epigenomic marks, including: acetylation marks H3K27ac and H3K9ac, associated with increased activation of enhancer and promoter regions” (DOI: 10.1038/nature14248); 3) “... H3K27me3 was shown to be highly enriched at the promoters of thousands of genes that are responsible for embryonic development and differentiation” (DOI: <https://doi.org/10.1038/emm.2017.11>).

In addition, irrespective of where eQTL are located in the genome, we believe that the enrichment or depletion of eQTL in the regions with distinct histone marks still provide useful information for the functional annotation of eQTL. We report the first eQTL map of the wheat genome and providing the research community information about the eQTL distribution across various genomic features identified so far in wheat is useful. Thus, we would prefer to keep the enrichment analysis without changes.

Comment 3. Authors showed that both cis- and trans-eQTL p-values positively correlate with the frequency of Hi-C contacts. A major point here is the resolution of the Hi-C data used that is not sufficient for a precise mapping of the interaction. In addition none of the interaction was validated by another method like FISH or 3C experiment. In addition to prove that some eQTL effects likely depend on the physical interaction between the regulatory elements and target genes, authors should at least demonstrate that some loops are destabilized in genetically diverse

allohexaploid wheat.

Response: We do agree with the reviewer that resolution of Hi-C analysis is limited and does not provide sufficient evidence for detecting interaction between individual eQTL and its target. For this reason, the results presented on Fig. 5e do require quite extensive experimental validation to be considered reliable. Because this specific analysis is not of central importance for main conclusions, we excluded this part of figure 5 as well as the related parts from the Results section of the revised manuscript. However, the results that were presented in the former Fig. 5a-d (now part of Fig. 3d-f) are intended to demonstrate global relationship between the distribution of eQTL and their effects among regions showing elevated frequency of chromatin contacts. For the purposes of these analyses, the resolution of Hi-C data is sufficient to demonstrate global trends or enrichment of eQTL within chromatin loops. For example, eQTL over-representation analysis within chromatin loops was performed by the human GTEx consortium (Aguet et al. Genetic effects on gene expression across human tissues. *Nature*. 2017;550:204–13). Their results demonstrated that eQTL are enriched within cis-regions: “However, similar to primary eVariant associations, secondary eVariants were enriched for chromosomal contact with target eGene promoters, as determined through Hi-C, compared to background variant–TSS pairs (Supplementary Information 6). This suggests that, despite their sequence-based distance from the TSS, primary and secondary eVariants are in close physical contact with their target gene promoters via chromatin looping interactions.”

We have modified Results section and retained only part of the Hi-C analyses, which are now presented as a part of Fig. 3d-f. In this section, we mostly focused on assessing the enrichment of eQTL within the regions of Hi-C contacts. We also indicated that 1) Hi-C data has limited resolution and could not be used for detecting the interaction of eQTL region with its target gene, and 2) that in our study it was used to check for enrichment of eQTL-target gene pairs within the regions involved in chromatin interaction. The Results section was modified as shown here: “Although, the low resolution of wheat Hi-C data does not allow us to map precisely regulatory regions involved in interaction, it could be used to assess enrichment of eQTL-target gene pairs relative to randomized data. First, we found that both cis- and trans-eQTL p-values positively correlate with the frequency of Hi-C contacts, suggesting that regions harboring eQTL-gene pairs showing strong association are also more likely to have higher frequency of chromatin contacts (Figs. 3d, 3e, Supplementary Table 7). Second, the regions harboring trans-eQTL between both homoeologous and non-homoeologous chromosomes showed elevated Hi-C contacts ($\log_{10}[\text{Hi-C}] = 1.24$) compared to a randomized distribution based on 100 samples (mean $\log_{10}[\text{Hi-C}] = 0.92$) (Fig. 3g). This result indicates that the probability of trans-eQTL-target gene pair occurrence within the regions forming chromatin loops is substantially higher than random. Among trans-eQTL-target gene pairs with a Hi-C contact frequency >50 , 15% were located with the homoeologous chromosome regions involved in chromatin interaction more frequently than non-syntenic regions (Fig. 3f)”.

And again, we would like to re-iterate that the goal of our study is to characterize genetic effects of variants on expression of duplicated genes in young polyploid genome and investigate their relationship to phenotype. Though comparison with chromatin states and chromatin loops was used in our study, the main purpose of these analyses was to confirm association of eQTL with functionally active region of the genome at the global scale. The direction of research that

reviewer is proposing is outside of the scope of our study, especially considering that the current manuscript is already quite complex. The other three reviewers suggested to simplify and shorten manuscript (please see response to comment 1).

Comment 4. Authors raised the possibility that the effects of trans-eQTL on target genes located in the homoeologous regions could be mediated by chromatin loops but again here this is supported by only the correlation between trans-eQTL and both homoeologous and non-homoeologous chromosomes. To my view this is not enough (see point 3)

Response: Please see our responses to comments 1 and 3. These analyses were never intended to be the main focus of the manuscript. As we noted there, we have excluded a section in Results specifically devoted to Hi-C analyses, including a specific case of overlap between eQTL and Hi-C (former Fig. 5e) from the manuscript. We have retained only those analyses that are based on the comparison of global patterns of overlap between eQTL and Hi-C, as a way of validating functional relevance of detected eQTL (now on Fig. 3d-f) and demonstrating that chromatin loops are enriched in eQTL-target gene pairs detected in our study. These approaches were commonly used in other eQTL studies before, including model systems (e.g. Aguet et al. Genetic effects on gene expression across human tissues. *Nature*. 2017;550:204–13), to investigate the functional relevance of detected eQTL.

Reviewer #2 (Remarks to the Author)

Major comments

Comment 1: The authors used a linear model method implemented in the Matrix eQTL software for eQTL detection. First of all, and most importantly, it is unclear how the cryptic relatedness in the sample was accounted for and the extent to which the eQTL test statistics were inflated due to the cryptic relatedness. It is also unclear whether three eigenvectors are sufficient to control for population stratification. The authors may need to demonstrate the distribution of the genomic inflation factors (GIF) for all the gene expression traits analyzed. One suggestion would be to use a linear mixed model method that can account for both relatedness and population stratification, as the authors did in GWAS analysis for the agronomic traits.

Response: We controlled population structure by providing three PCs during the calculation of PEERs that was used for removing the effects of confounding factors, including the effect of population structure, on gene expression levels. The PEER residuals were then used in association mapping with Matrix eQTL followed by calculating the corresponding FDRs using BH correction. Overall, we applied the eQTL mapping procedure, which was used by the humans Genotype-Tissue Expression consortium (doi:10.1038/nature24277) for mapping eQTL and shown to produce adequate results with low false positive rate. Based on the estimates of genomic inflation factor (GIF), this approach was effective in controlling population structure for nearly 61% of genes, which showed no evidence of inflation of test statistics (61% of genes had

GIF<1.1). Per reviewers suggestion, we have included the distribution of GIF in our dataset (Supplementary Fig. 15).

These results indicate that for the majority of genes, three PCs are sufficient for controlling the effect of population structure and does not require including relatedness matrix. While the remaining genes do show some effect of population structure on test statistic, but these effects are not dramatic and often observed in many GWAS studies. Selecting the optimal combination of PC numbers and/or relatedness matrix for inclusion into the model for every individual gene in eQTL mapping studies remains computationally challenging, and to our best knowledge was not performed in most of the eQTL studies in humans or model systems. For this reason, to reduce the chance of detecting false positive SNP-trait association, we have applied strict FDR correction ($FDR < 10^{-5}$). The effect of this FDR threshold on false positive rate has been evaluated by permutation (now described in the Methods). For more detailed response related to permutation and applied p-value threshold please see the response to comment 2 below.

Comment 2: Another critical issue is the choice of statistical significance level for claiming eQTLs. The authors chose an FDR threshold of $1e-5$ without justification. However, the number of false positives in the reported eQTLs is unclear, especially considering that relatedness has not been accounted for in the eQTL analysis.

Response: The effect of the selected $FDR = 10^{-5}$ on proportion of false positive eQTL was investigated by performing permutation analysis. The expression values of each of the 52,060 genes in our dataset were permuted relative to genotyping data (includes 2,021,937 SNPs) to generate 1000 randomized datasets. The SNP-gene expression association test statistic was calculated using the same approach described in the paper. By applying p-value threshold corresponding to $FDR \leq 10^{-5}$, on average, we detected 3,595 associations in the randomized datasets. In the real-life dataset, we have identified 11,421,859 associations passing this threshold indicating that only 3.2×10^{-4} associations passing our threshold are false positives. The false positive rate calculated using the permuted data for spikes resulted in 0.8×10^{-3} . This number of false positive associations suggest that the selected FDR thresholds and approaches used for eQTL mapping will unlikely result in elevated rate of false positives and will have major impact on the reported results. In the Methods section we provided detailed description of the performed permutation analyses.

Comment 3: This is reflected by the large number of eQTLs detected (after LD-based merging) from both the seedling and spike data sets, irrespective of the small sample sizes. Although the sample size of the seedling data is more than twice that of the spike data, the number of eQTLs detected from the former (36,898) is substantially smaller than the latter (65,117). Interestingly, after integrating the eQTL data with the GWAS, the number of trait-associated genes using the seedling eQTL data (329) was much larger than that using the spike eQTL data (95), consistent with the difference in sample size between the two data sets.

Response: Thank you for pointing out at this unexpected number of eQTL reported for spikes compared to those reported for seedlings. This helped us to identify a problem with data processing and fix it. Overall, the excessive number of reported spike eQTL was not due to

elevated false positive rate, but due to issues related to the LD-based clustering of SNPs significantly associated with variation in gene expression.

We have assessed the proportion false positives in eQTL mapping results for spikes by performing eQTL mapping using permuted data. We have generated 1000 randomized datasets and applied p-value threshold corresponding to $FDR \leq 10^{-5}$. While in the original non-permuted dataset, 1,336,626 SNPs pass this p-value threshold (before LD merging), in the permuted datasets, on average we had only 10,858 SNPs passing threshold, suggesting that in our experiment the false positive rate is around 0.8×10^{-3} . This rate is higher than in the seedling dataset (3.2×10^{-4} ; see response to comment 2 above), however, it is well within the acceptable range for QTL mapping studies.

Our further analysis of spike eQTL showed that the increased number of reported variants was not due to elevated number of false positives, as was suggested by the reviewer, but due to the reporting redundant set of SNPs that were not properly clustered based on the physical proximity or LD. We found that the LD- and distance-based clumping was not performed using the same criteria we applied for eQTL from seedling tissues. By applying the same criteria, we have obtained a set of 15,238 eQTL. This number of eQTL is consistent with the smaller size of the population used for eQTL mapping in spikes compared to population sies used for mapping in seedlings. The Result section has been updated and updated eQTL detected in spikes are provided in Supplementary Table 5.

Comment 4: Several observations might indicate that there are a substantial number of false-positive trans-eQTLs. 1) One observation is related to the expression correlation. If genetic variations are expected to disrupt the expression correlation between homoeologous genes, then the expression correlation between homoeologous gene pairs that share eQTLs is expected to be smaller than that for random gene pairs (Figure 4). This is the case for genes that share cis-eQTLs but not for those that share trans-eQTLs. On the other hand, the bimodal distribution (e.g., the mode on the right-hand side) may indicate the enrichment of false positives.

Response: It was not very clear to the authors why this reviewer assumed that the information shown on Fig. 4 suggests that we have false positive eQTL? It is likely coming from some not clearly formulated statements in the manuscript, and we do apologize for that. It is critical to note that cis-eQTL by definition could not be shared between homoeologs because cis-eQTL is linked only with the gene that is on the same chromosome, whereas homoeologs by definition are genes on different homoeologous chromosomes. Therefore, only trans-eQTL eQTL could be shared between homoeologs, and statement in the reviewer's comment "This is the case for genes that share cis-eQTLs but not for those that share trans-eQTLs" could not be applied to data we reported in the manuscript.

When we talk about disruptions of homoeologous gene regulation, we specifically talk only about *cis*-eQTL. The results on Fig. 4c are based on shared trans-eQTL, which appear do not disrupt correlation of expression. There are several pieces of data shown in the manuscript that explicitly link dysregulation (homoeolog expression bias) with *cis*-eQTL: 1) Figure 2d shows that density of SNPs around the gene (gene body +/- 10 kb) and correlation in the expression of homoeologs, declines with increase in the density of SNPs consistent with the presence of cis-regulatory SNPs

as diversity around genes goes up. 2) Fig. 4e (third panel – Configuration 3) shows that presence of cis-eQTL in homoeologs (these cis-eQTL are not shared between homeologs because each regulate only corresponding homoelog in cis-configuration) shifts correlation to lower values with the mean SCC = 0.17.

The reviewer likely points at Fig. 4c, where correlation between homeologs increases with increase in the number of shared eQTL. However, in this figure we present the effect of shared trans-eQTL, and these results have nothing to do with cis-eQTL, which indeed dysregulate homeologs. In our study we did not find evidence that trans-eQTL dysregulating homeologs, as was shown on Fig.4d. In addition, the analysis of trans-eQTL shown on Fig. 4e (Configuration 2) also shows that homeologs affected by these trans-eQTL show high levels of expression correlation as evidenced by high mean SCC = 0.6 and presence of second density peak with the mean SCC ~ 0.8.

Comment 5: 2) The second observation is related to overlap between Hi-C contacts and trans-eQTL-target gene pairs. As presented in Figure 5d, the density plot for trans-eQTL-target gene pairs is highly similar to that for a random set of genes. BTW, I do not think the test has been done correctly in this case. A proper test should be a test of the mean (mode or median) of the number of Hi-C contacts for the trans-eQTL-target gene pair against the distribution of the mean (mode or median) values obtained from repeated sampling of random gene sets (rather than one random gene set).

Response: We compared the overlap of trans-eQTL-target gene pairs between observed and randomized data as suggested by the reviewer. The results clearly indicated a substantially higher level of chromatin interaction between the regions harboring the trans-eQTL-target gene pairs compared to chromatin interaction in the randomized data (calculated from 100 random samples). The results are updated and now presented on Fig. 3g, and the respective section of the results.

The Results section is modified as shown here: “Second, the regions harboring trans-eQTL between both homoeologous and non-homoeologous chromosomes showed elevated Hi-C contacts ($\log_{10}[\text{Hi-C}] = 1.24$) compared to a randomized distribution based on 100 samples (mean $\log_{10}[\text{Hi-C}] = 0.92$) (Fig. 3g). This result indicates that the probability of trans-eQTL-target gene pair occurrence within the regions forming chromatin loops is substantially higher than random.”

Comment 6: 3) The third observation is the lack of overlap between the trans-eQTLs and the QTLs for the agronomic traits.

Response: We believe that the reviewer mis-interpreted the reported results for overlap between trans-eQTL and eQTL for agronomic traits. The lack of enrichment (as was reported in the manuscript) does not mean that there is no overlap, and does not indicate that *trans*-eQTL detected in our study are false. Using even quite strict criteria for overlap, we observe 36 marker-trait associations for major agronomic traits overlapping with trans-eQTL. We simply do not observe enrichment, indicating that for most traits variation is preferentially controlled by cis-regulatory variants rather than by trans-regulatory variants. These results are consistent with

other studies performed in humans. For example, in an early eQTL study in humans, it was shown that SNPs associated with Crohn's disease show enrichment for cis-eQTL, but do not show enrichment for trans-eQTL: "The enrichment was preserved in the SNPs classified as cis-regulators but was not evident in the SNPs classified as trans-regulators, suggesting that cis-regulatory effects were more likely to be present among the Crohn's associated SNPs" (cited from 10.1371/journal.pgen.1000888). We have added a sentence to the Results section interpreting these results: "Consistent with an earlier study, these results suggest that the trait-associated SNPs are more likely to be cis-regulatory rather than trans-regulatory variants".

Comment 7: 4) The proportion of genes with trans-eQTLs that also have cis-eQTLs is surprisingly low, especially in the seedling data (1469 / 8315). Is it also an indication of the potentially elevated false-positive rate?

Response: Based on the permutation analysis (see responses to previous comments), we have established that our study does not suffer from extremely high false positive rate. Thus, we do not believe that the reported overlap between *cis*- and *trans*-eQTL is an indication of elevated false positive rate.

Theoretically, every *trans*-eQTL could also be a *cis*-eQTL for some genes that act as regulators of distant targets. However, based on our knowledge, this is not the case in any eQTL study so far published. For example, in a study performed in humans, only 18.5% of *trans*-eQTL were also declared as *cis*-eQTL (doi: 10.1371/journal.pgen.1004461). The overlap between *cis*- and *trans*-eQTL is mostly defined by the parameters used to classify eQTL into *cis*- and *trans*-acting variants, and also by the genomic distribution of SNPs, rate of LD decay etc. Wheat is a young (10,000 years old) hexaploid species with huge genome (17 Gb, or ~5.6 Gb per constituent genome). It experienced recent domestication bottleneck leading to low overall genetic diversity, especially in the D genome. Nearly 2/3 of each chromosome arm show extremely low recombination rate and slow rate of LD decay with LD blocks spanning tens of megabases of DNA. As result of these factors, many eQTL could be located far away from their associated genes. In our study, we defined *cis*-eQTL as those that are located within 2-Mb window around associated genes. The *trans*-eQTL were declared only when they are associated with the expression of genes on other chromosomes (the approach often used in humans eQTL studies; for example in Aguet et al. Genetic effects on gene expression across human tissues. *Nature*. 2017;550:204–13). Considering these factors, it is possible to have many *trans*-eQTL that are not *cis*-eQTL because they are located outside of the 2 Mb boundary from genes. Thus, we believe that the reported overlap between *cis*- and *trans*-eQTL is not something alarming and certainly is not an indicator of high false positive rate (as was demonstrated by the permutation analyses).

Comment 8: Ln320-321. The observed negative correlation could be due to ascertainment. The statistical power to detect an eQTL is a function of $n * 2p(1-p) * b^2$ with n being the sample size, f being the allele frequency and b being the effect size so that the effect size of a lower MAF variant needs to be larger to be detected at a specific significance level.

Response: Thank you for pointing out at this relationship of which we were aware from the published eQTL studies in other species. However, we should note, that all these studies indicate

that even though decreased power to detect rare eQTL could contribute to the observed trend, it does not account for all observed negative relationship between the allele frequency and effect size (doi: <https://doi.org/10.1534/genetics.118.301833>). Our results are consistent with prior findings. One of the approaches to assess the effect of selection on eQTL is to investigate eQTL with relatively large effect sizes that could be detected at all frequencies. We used eQTL that had effect sizes equal to or greater than the minimum effect of eQTL at frequency equal to or less than 0.05. We end up with 82 cis-eQTL for singletons and 2,324 cis-eQTL for homoeologous gene triplets. The analysis of correlation between MAF and effect size for homoeologous triplets showed significant negative relationship (p-value = 0.001). Similar analysis performed using single copy genes showed lack of significant correlation. However, we attribute the lack of correlation in this case to small sample size for singleton cis-eQTL that increase contribution of rare large-effect QTL on the distribution of effect sizes across different MAF classes. Thus, this analysis using the filtered data does not exclude the possibility that *cis*-eQTL of the homoeologous triplets are under selection, in spite of genetic redundancy created by polyploidization. We have updated Results section and added Supplementary Fig. 7 to incorporate these analyses into the manuscript.

“The negative relationship between frequency and effect size was observed in homoeologs even for the subset of *cis*-eQTL whose effects are detectable at all frequencies (Supplementary Fig. 7)”.

Comment 9: Ln855-857 and Ln887-889. The authors may need to clarify the purpose of using BLUEs and BLUPs of the phenotypes for follow-up analyses and why they used BLUE for the 800 accessions and BLUP for the 400 accessions.

Response: BLUEs were taken from the previously published study (<https://doi.org/10.1038/s41588-019-0382-2>). While in the current study we used BLUPs, we also estimated BLUEs for 400 accessions and showed that correlation b/w BLUEs and BLUPs is 0.994, suggesting that both sets of values are similar.

Comment 10: Ln956 “Ten-fold cross-validation”. This method is only applicable to data where all the individuals are independent.

Response: The panel used in the study is composed of unrelated accessions from worldwide collection of lines, which makes the subsets of individuals from the panel used as training dataset and testing dataset being independent of each other. This technique is commonly used for testing the phenotypic prediction accuracy in breeding. The implementation of cross validation approach used in our study is similar to one used for predicting traits in maize using expression values of 5,000 genes (<http://dx.doi.org/10.1038/nature25966>). We have added the reference to this study in the Methods section.

Minor comments

Comment 11: Ln197 “Only half of these genes”. Please be specific about “these genes”.

Response: Thank you for noting this. Corrected as follows: “Only 47.8% of these 6,173 genes were located in the D genome...”.

Comment 12: Figure 2c. The authors may need to clarify how this plot was made because genetic variance is estimated for a single gene, whereas expression correlation is computed for a pair of genes.

Response: The plot was prepared by calculating correlation coefficients (SCC) for all possible pairs of homoeologs and plotting it against the proportion of genetic variance for these homoeologs. The mean and standard error of genetic variance was calculated for data binned based on the ranked SCC values. The figure legend was updated.

Comment 13: Ln560 Fig. 7f?

Response: We have excluded this figure from the revised manuscript. This analyses require more detailed description and presentation in the manuscript, and we decided to focus this last section of the manuscript on testing the association between homeolog expression bias and phenotypes.

Reviewer #3 (Remarks to the Author):

The authors investigated the regulatory control of duplicated gene expression in hexaploid bread wheat from a popgen perspective and its relevance to agronomic traits:

Comment 1: 1. partitioning genetic variation of expression traits to evaluate regulatory control from the same (cis-acting variants) or different (trans-acting) subgenomes, revealing stronger effects from former. Line 244 concluded that "dysregulation of homoeologs is primarily associated with the cis-regulatory diversity": what does "cis-regulatory diversity" refer to and how is it connected with the cis-acting variants ?

Response: We used these two terms interchangeably in manuscript. Both terms refer to cis-eQTL that have regulatory effect on the expression of target genes. The usage of this terminology appear to be common, and for example was used with the same meaning by the human GTEx consortium (Aguet et al. Genetic effects on gene expression across human tissues. *Nature*. 2017;550:204–13).

Comment 2: So there are A LOT of good ideas and tests involved, and I do believe the authors are onto some quite novel aspects on the regulatory complexity in allopolyploid plants, but also because of such complexity, this manuscript is not easy to follow. The title is rather lengthy but doesn't have a clear point. The abstract appears to cover main results, but these results are disconnected to deliver a cohesive story.

One major hurdle for me to understand this manuscript is the concept of "homoeolog dysregulation". According to Figure 1a, the authors used this term to describe homoeologs that are differentially expressed, which doesn't indicate any functional consequences like impairment in protein functions or metabolic process. In that case, there are already terms like "unequal expression" and "homoeolog expression bias" for that. But later, "dysregulation" was defined more specifically based on negative SCC. All these terminological and conceptual inconsistency brought by the new term make it difficult to connect this work with relevant literatures to comprehend new evidence and findings.

Response: In the revised manuscript, we decided to stick with previously used terminology that appear to be sufficient to describe our results. To describe unequal expression of homoeologs at the individual accession's level, we used "homoeolog expression bias" or "homoeologs with biased expression". We used "negatively correlated homoeologs" term to describe cases of homeologous gene pairs where increased frequency of accessions with biased gene expression leads to negative expression correlation in the panel.

Comment: Another issue is the use of cis and trans, more detailed comments are marked in the attached PDF.

Remark: Below, starting from Comment 3, we have provided detailed responses to the comments from the PDF file.

Comment 3: Methods seemed sound to me and carefully executed for each task. For example, I do appreciate the efforts the authors put in to assess transcript estimation, which employed simulated RNA-seq data to validate the high correlation between observed and expected reads.

Response: Thank you.

Comment 4: Lengthy title hard to grasp the key finding, possible to make it concise?

Response: We tried to shorten title and make it more informative. The current title is "The genetic architecture of homoeologous gene expression dosage variation and its impact on agronomic traits in allopolyploid wheat".

Comment 5: Do you mean to test this hypothesis, or is this a finding? This is related to L.45-48 in Abstract "We hypothesize that these cis-acting variants have likely been exploited for improving productivity traits in wheat, and depending on their effects on component phenotypes, were either purged from or accumulated in the population."

Response: The abstract was revised for clarity. For example, this sentence now reads as follows: "We hypothesize that the frequency of these cis-acting variants has likely been impacted by breeding for increased productivity."

Comment 6 related to L. 74-75: “ In addition, many regulatory variants linked with one of the homeologs have the potential to dysregulate a gene’s expression and change its dosage relative to other homeologs (Fig. 1a).” how to define dysregulation? does it have any functional consequences or simply refer to changes in expression levels ?

Response: We would like to thank reviewer for raising concerns related to the terminology used to describe the patterns of expression we observed in the wheat panels. During writing the manuscript, we also felt that the usage of “homeolog dysregulation” term might be confusing and also complicate connecting our findings to existing body of work on regulation of polyploid gene expression. In the revised manuscript, we decided to stick with previously used terminology that appear to be sufficient to describe our results. To describe unequal expression of homeologs at the individual accession level, we used “homeolog expression bias” or “homeologs with biased expression”. We used “negatively correlated homeologs” term to describe cases of homeologous gene pairs where increased frequency of accessions with biased gene expression leads to negative expression correlation in the panel.

Comment 7 to L. 109-110: all bread wheat?

Response: Yes, all are bread wheat accessions. We modified the sentence to indicate that.

Comment 8 to L. 115: “A simulation-based estimate suggested 98% correct read mapping (see Methods).” After reading the method section, it kinda makes sense to me. But this sentence here needs clarification, for example, what is the purpose of such test, and how does this simulation result validate the mapping of real data.

Response: Such kind of simulations have been extensively applied in studies involving analyses of NGS data. Usually this was done to assess the impact of software settings on the accuracy of read mapping. For these reason we did not provide detailed description of reasons why we performed them assuming that readers are well familiar with this approach. In many our prior genomic studies we used simulations to show that we can accurately map reads to distinct subgenomes. Because simulated data is derived from real-life genomic data by subsampling, it is not that different with regard to the quality of reads and their distribution across genome compared to real-life data. The advantage of using simulated data is that simulated reads are randomly selected from the genomic regions with known locations within the reference sequence. As a result, we could assess the accuracy of read mapping because we know where each read should be mapped within the reference.

We modified sentence: “A simulation-based estimate suggested that the alignment settings used in our study provide 98% correct read mapping to the polyploid wheat genome”.

Comment 9 to L. 117: what do these two gene numbers refer to, transcripts and gene loci?

Response: Thank you for noticing this error. Another reviewer also pointed at inconsistency of these numbers and those that are reported in the Methods section. We amended this sentence as

follows: “Expression levels measured as Transcripts Per Million (TPM) were estimated for high-confidence (HC) genes in RefSeq v.1.0, with 52,511 transcripts (47,274 genes)....”

Comment 9 to L. 143 (Fig. 1c legend) and L. 161-163: Do you mean if the expressions are 3,4,0.1 for a triplet, the X and Y values are $(3+4+0.1)/3$ and $(3+4)/2$, respectively? Or do you mean the 198 means of a triplet were split into two different groups, with one group met the above two conditions and the other group didn't? In either case, this sentence and figure legend need to be rephrased to be clear.

Response: We tried to clarify the method of analysis used for generating Fig. 1c. The text was modified as follows: “...we selected a set of 1,443 gene triplets that met two criteria: 1) one out of three homoeologs was downregulated ($TPM < 0.1$) in at least two wheat lines, and 2) at least two wheat lines have all three homoeologs expressed ($TPM > 2$). We applied this criteria to each triplet to split 198 accessions into two groups, one group composed of accessions with one of the homoeologs downregulated and another group including accessions with all three homoeologs expressed. The sum of expression values from all three homoeologs ($A+B+D$) was calculated for each accession and used to derive the mean of total homoeologs' expression for each group. . In most cases, the mean expression ratio between these two groups across gene triplets (Fig. 1d) was below 1:1,....”

The Fig. 1c legend was amended as follows: “The mean of the sum of the total triplet expression ($A+B+D$) in groups of accessions with (y-axis) and without (x-axis) one of the gene copies downregulated.”

Comment 10 related to L. 187 which refers to Suppl. Figs. 2a and 2b. Is this correct? They don't seem to compare cis and trans.

Response: Thank you for noticing this error. The reference to Suppl. Figs. 2a and 2b is wrong and it was removed from this sentence.

Comment 11 related to L. 196-197 “This observation is likely associated with the lack of cis-regulatory diversity around these genes (Fig. 2b).”: Very interesting. What about the % of total variance can be explained between genes mostly explained by cis and by trans? If comparable, that will suggest that trans effects are truly dominant and do not require cis diversity. If $trans < cis$, that means the lack of cis diversity contributes to weak cis effect, while trans alone cannot explain enough variance.

Response: As suggested, we have performed additional analyses and added information about the mean total expression variance explained for genes whose expression is mostly explained by cis-genic (SNPs 2 Mb around genes) or by trans-genic (SNPs from outside of 10 Mb region around genes and also located on all other wheat chromosomes) variants. The means in these two groups genes were similar (47.0% vs. 46.4%).

The following paragraph was added: “We further split all SNPs into cis-genic (SNPs within +/- 1 Mb region around genes) and trans-genic (SNPs outside of +/- 5 Mb region around genes and SNPs located on other wheat chromosomes) subsets. After partitioning variance for gene expression, we compared the means of total variance explained for two sets of genes: 1) genes with variance explained only by cis-genic SNP, where trans-genic SNPs contribute <1% to variance, and 2) genes with variance explained only trans-genic SNPs, where cis-genic SNPs contribute <1% to variance. The means in these two groups of genes were similar (47.0% vs. 46.4%), suggesting that in the cases of cis-genic diversity loss, the contribution of trans-genic effects in allopolyploid genome to expression variance could be similar to the contribution of cis-genic effects.”

This comment also prompted us to rethink the results of analyses presented in Fig. 2c, where we plotted the total proportion of gene expression variance for individual homeologs and correlation coefficients between the pairs of homeologs. We have re-analyzed gene expression variation data using SNPs that were partitioned into cis-genic and trans-genic sets as described above. In the updated Fig. 2c, we plotted the proportions of genetic variance explained by these two groups of SNPs. This analysis allowed us to clearly separate cis-effects from trans-effects, and shows that negative or lack of expression correlation between homeologs is associated with increased proportion of *cis*-acting variants. The contribution of *trans*-acting variants to homeolog expression correlation was more or less inform across different expression correlation levels. We observed increase in the contribution of *trans*-acting variants relative to contribution *cis*-acting variants to gene expression variation with increase in the levels of expression correlation, which is consistent with the results of eQTL that were presented later in the manuscript. For example, Fig. 3c shows that increase in the number of shared eQTL between homeologous gene pairs coincides with the increase in the expression correlation levels.

The figure 2c legend was revised as follows: “The relationship between the proportion of genetic variance explained by cis- and trans-genic SNPs calculated for individual homeologs and the levels of expression correlation (SCC) between the pairs of homeologs in the wheat panel. The mean of genetic variance was calculated for data binned based on the SCC values.”

The following modifications are made in the Results: 1) Section title changed to “The effects of cis- and trans-acting variants on expression correlation between homeologs” ; 2) First paragraph of this section is modified as follows: “The combined effect of *trans*- and *cis*-acting variants on individual homeologs defines the relative expression of homeologs. We compared the proportions of expression variance in individual homeologs explained by *cis*- and *trans*-genic SNPs among homeologous gene pairs showing distinct levels of expression correlation (SCC) (Fig. 2c). An increase in SCC was accompanied by a decrease in the total expression variance explained, with the largest proportion of explained variance observed for homeologs showing $SCC < 0$ (Fig. 2c). While SCC increase was accompanied by 3-fold decrease in variance explained by *trans*-genic SNPs, more substantial 11-fold decrease in explained variance was observed for *cis*-genic SNPs, reaching 1.6% for homeologs showing high correlation in the expression levels ($SCC > 0.90$) (Fig. 2c). These results suggest that discordant expression of homeologs is largely associated with the presence of *cis*-acting variants in homeologous genes.”

Comment 12 to L. 207-212 (legend for 2a): Description for the seedling panel? It need to be clarified, maybe adding "As shown in the seedling panel"

Response: The figure legend was modified as suggested.

Comment 13 to L. 223-224 "... we examined the relationship between the variance in homoeolog expression explained by genome-wide SNPs (Fig. 2a) and the levels of expression correlation (SCC)...": In fig2a, variance explained by genome-wide SNPs is for each gene (homoeolog), correct? For a pair of homoeologs, how is the variance determined to be examined with SCC? Or do you indicate some measure of genetic difference between homoeologs?

Response: We did not specifically estimated the genetic variance for difference in gene expression between homeologs. Along the y-axis we plotted the genetic variance calculated for individual homoeologs from pairs used to calculate correlation coefficients (in other words, for each measure of SCC on the x-axis of the plot we had the values of genetic variance for two homoeologs on the y-axis). We have modified the text to clarify this point.

Comment 14 to L. 231-233 "This conclusion is consistent with a decrease in the number of common and rare SNPs around the homoeologs with an increase in their levels of expression correlation (Fig. 2d)": I don't understand the reasoning here. Do the numbers of SNPs represent cis divergence between homoeologs? So lower cis diversity could contribute to higher SCC between homoeologs? Then isn't it in conflict with the next sentence suggesting "no or weak relationship"? Is it possible to directly partition cis and trans contributions to this negative correlation? Also what is the difference between common and rare alleles?

Response: Variants that distinguish two genomes (and homoeologs) from each other are divergent sites inherited from diploid ancestors, and cannot be referred to as "SNPs". In this case, we considered common and rare SNPs that are true polymorphisms segregating within individual genomes. We looked for the total number of SNPs at and around homoeologs (+/- 10 kb) and found that the decrease in SNP number coincides with increase the level of expression correlation. This conclusion is also consistent with the new analysis, were we partitioned expression levels of individual homoeologs using cis-genic (+/- 1 Mb around genes) and trans-genic (SNPs 10 Mb outside of genes plus SNPs from all other chromosomes) SNPs (Fig. 2c). We could see also that the contribution of cis-genic SNPs for expression variance of individual homoeologs decrease in homoeologous gene pairs that show the high levels of expression correlation.

We apologize for unclearly formulated statements in the next sentence that refers to Supplementary Fig. 3. We wanted to note that this negative correlation between the number of rare/common SNPs and SCC persists even if SNPs from different genomes are analyzed separately. The main reason for repeating this analysis for genome-specific sets of SNPs is that in wheat D genome shows dramatically lower level of diversity (due to severe population bottleneck associated with polyploidization) compared to the A and B genomes, and we wanted

to see if this negative trend between diversity and SCC could be found in distinct genomes (Supplementary Fig. 3a and 3b). We indeed found that this trend is consistent across all three genome, and in that sense, negative correlation is independent of the overall genome-specific levels of genetic diversity and related to local cis-diversity around every gene.

The testing of the dependence between inter-homoeolog divergence and SCC was performed to assess to what extent the levels of expression correlation between homoeologs are defined by the levels of sequence divergence between the wheat genomes. If we would observe the negative correlation between inter-homoeolog divergence and SCC, similar to the one observed between cis-diversity and SCC (Fig. 2d), then we would have to attribute this negative correlation to inter-genomic divergence at the regulatory regions that came from the diploid ancestors. However, we did not observe strong negative correlation divergence and SCC, and therefore, our assumption that local cis-diversity is the main factor driving expression correlation between homeologs still holds.

To clarify these points, we have modified the text in this paragraph as follows: “These trends were consistent across all three wheat genomes (Supplementary Fig. 3a and 3b), indicating that negative relationship between cis-diversity and SCC is independent of genome-specific levels of genetic diversity. The lack of strong relationship between the inter-homoeolog sequence divergence and SCC suggests that inter-genomic divergence at the regulatory regions does not have global effect on homoeolog co-expression (Supplementary Fig. 3c)”.

We have indicated in the text the criteria used to separate SNPs into rare ($MAF < 0.05$) and common ($MAF \geq 0.05$) based on their population frequency. The rationale for splitting variants into these two groups is grounded on previous studies, which showed that rare variants around genes could have much stronger cumulative effect on expression variation, often resulting in extreme deviation from population mean expression, compared to more common alleles.

Comment 15 to L. 237-238 “... from lowest to highest expression levels across wheat accessions, and the rare allele load in the upstream...”: Not clear to me what is the purpose to analyze rare allele load.

Response: As was mentioned in the response to the previous comment, the cumulative effect of rare alleles ($MAF < 0.05$) on gene expression variation could be much higher than the effects of common alleles (for example, see ref. Kremling et al. *Nature*. 2018;555:520–3, <http://dx.doi.org/10.1038/nature25966>). Quantifying rare allele load in the upstream promoter regions and relating it to gene expression levels is one of the methods for assessing the effect of rare allele load on gene expression levels. By performing this analysis we wanted to assess the possible impact of rare allele load (excluded due to low frequency from the eQTL mapping) on the level of homeolog expression correlation.

We modified sentence : “To assess the impact of rare *cis*-variants on the homoeolog expression levels, we investigated the relationship between the expression ranks of each homoeolog....”.

Comment 16 to L. 244-245 “...the dysregulation of homoeologs (Fig. 1a) is primarily associated with the cis-regulatory diversity.”: So far I can understand that differential expression of homoeologs can be attributed to genetic variance from the same subgenome.

Response: Based on the results of new analyses performed in response to this reviewer’s comments and presented on Fig. 2c and 2d, and described in the corresponding sections on Results, our conclusion still holds and we do observe association of differential homeolog expression with the accumulation of *cis*-variants.

Comment 17 to L. 252-254 “A conservative criterion was applied to define trans-eQTL as eQTL located in different genomes or chromosomes relative to the target gene, and cis-eQTL as eQTL located within 2 Mb of a target gene. According”: Was distant cis-eQTL excluded from following analysis, and why?

Response: Because it is usually difficult to clearly separate true distant *cis*-eQTL from true *trans*-eQTL. The recombination rate in wheat, overall quite low compared to many other species, also varies dramatically across large 16 Gb-genome, where recombination is severely suppressed in pericentromeric 2/3 of each chromosome arm. As a result it is quite complicated to come up with fail-proof criteria to separate distant *cis*-eQTL from *trans*-eQTL. As a result, we opted to use conservative approach to define *cis*- and *trans*-eQTL. Similar approach for defining *cis*- and *trans*-eQTL was previously used by the Human GTEx consortium (see ref. Aguet et al. Genetic effects on gene expression across human tissues. *Nature*. 2017;550:204–13). Besides, based on the criteria used in the large maize genome to define distant *cis*-regulatory elements, most of which overlap with eQTL, are located less than 1 Mb away from a gene (see Fig 1c, 1d in ref with DOI: 10.1038/s41477-019-0547-0), we should have already included many distant *cis*-regulatory region within the analyzed 2 Mb region around genes.

Comment 18 to L. 286 in the legend of Fig. 3: What about MSF vs MRF, the differential enrichment test? How is different from MSF and MRF test?

Response: We have added additional explanation to the legend of figure 3: “Enrichment was assessed relative to genome-wide patterns, except for MSF vs MRF, where enrichment was tested for eQTL located within MSF relative to MRF.”

Comment 19 to L. 700 “... as well as de novo, assembled transcripts were combined...”: Part of RefSeq v 1.0 or elsewhere? Please clarify.

Response: In the revised manuscript, we have excluded mentioning of de novo assembled transcript data. Though we have assembled unmapped reads, they have not been used for eQTL mapping or included into any other analysis.

Comment 20 to L. 823-824 ” eQTL located more than 2 Mbp away from an eGene were used to define distant cis-eQTL.”: This sentence needs work.

Response: Thank you. This sentence should have been removed even from the original submission of the manuscript. It appears that mentioning about this group of eQTL was left in

the Methods. Based on the reasons provided in response to an earlier comment (comment 17) from this reviewer, due to difficulty of separating distant *cis*-eQTL from *trans*-eQTL, we end up not using this group of *cis*-eQTL in our analyses. We removed this sentence from the Methods section.

Reviewer #4 (Remarks to the Author)

Comment 1: Previous work has identified *cis* and *trans* eQTLs polyploid species including polyploid *Arabidopsis* (Shi et al. 2012), potato (Zhang et al. 2020) and cotton (Bao et al. 2019) but this study would be the first to identify them in wheat and study how they interact to affect homoeolog expression. The study also attempts to associate dysregulated expression of homoeologs with trait values. However, this dysregulation-trait association is less robust compared to the genetic partitioning and eQTL analyses.

Bao Y, Hu G, Grover CE, Conover J, Yuan D, Wendel JF. Unraveling *cis* and *trans* regulatory evolution during cotton domestication. *Nature communications*. 2019 Nov 27;10(1):1-2.

Zhang L, Yu Y, Shi T, Kou M, Sun J, Xu T, Li Q, Wu S, Cao Q, Hou W, Li Z. Genome-wide analysis of expression quantitative trait loci (eQTLs) reveals the regulatory architecture of gene expression variation in the storage roots of sweet potato. *Horticulture Research*. 2020 Jun 1;7(1):1-2.

Shi X, Ng DW, Zhang C, Comai L, Ye W, Chen ZJ. *Cis*- and *trans*-regulatory divergence between progenitor species determines gene-expression novelty in *Arabidopsis* allopolyploids. *Nature communications*. 2012 Jul 17;3(1):1-9.

Response: Thank you for suggesting these references. One of them (Bao et al., 2019) was already cited in the manuscript. However, we would like to note that only Zhang et al., 2020 study actually mapped the genomic location of eQTL. The remaining two studies investigated *cis*- and *trans*-regulatory effects using F1 hybrids, but did not perform eQTL mapping. Out of these two studies we cited Bao et al., 2019 study because it was performed in natural allopolyploid species. While the focus of Shi et al., 2012 study on newly synthesized allopolyploid is very interesting, the processes occurring in these new allopolyploids are more related to regulatory changes at the early phases of evolution after polyploidization, rather than to regulation of genes in established polyploids. We cited all three papers in the Introduction.

Major Comments

Comment 2: Is the term “balanced” accurate for what is described in terms of homoeolog expression? Balanced in previous work (e.g. Ramirez-Gonzalez et al., 2018, *Science*) refers to similar expression levels of A, B and D homoeologs e.g. all three are expressed at 1 TPM. My understanding is that here by “balanced” the authors mean that the expression level of the homoeologs is correlated but not necessarily at the same expression level e.g. if the ratio between

A:B:D is 4 TPM: 1 TPM: 1 TPM and that is consistent across the population (SCC will be positive) the relationship is considered “balanced”. This seems non-intuitive based on the terminology. Would “positively correlated homeologs” be a more suitable term to avoid confusion with previous work in this area? The graphic in Fig 1A should be updated using multiple possible relationships: currently the nuance about the correlated expression is lost when relying on that figure for definitions. (i.e. include a line at 1:1 and a line at e.g. 3:1 to show the expression level of A:B does not have to be equal).

Response: Thank you for comments on the usage of terminology. In response to comment 6 from reviewer 3, we have modified manuscript. During writing the manuscript, we also felt that the usage of “homeolog dysregulation” term might be confusing and also complicate connecting our findings to existing body of work on regulation of polyploid gene expression. In the revised manuscript, we decided to stick with previously used terminology that appear to be sufficient to describe our results. To describe unequal expression of homeologs at the individual accession level, we used “homeolog expression bias” or “homeologs with biased expression”. We used “negatively correlated homeologs” term to describe cases of homeologous gene pairs where increased frequency of accessions with biased gene expression leads to negative expression correlation in the panel.

The term “balanced” expression is not specific to wheat and was used in early studies of aneuploid and polyploid plants of different ploidy level (for review see Birchler and Veitia, The gene balance hypothesis: from classical genetics to modern genomics. *Plant Cell*. 2007;19:395–402). This term is also used to describe the relative expression dosage of tandemly duplicated paralogs (10.1126/science.aad8411). In relation to early studies of homeolog expression in allopolyploid wheat, indeed balanced expression was referred to 1:1:1 expression of the A, B and D genome homeologs, with some deviation from this ratio. However, these early studies are focused on a single accession of wheat, which does not allow to assess the actual population mean of the expression of each homeolog. In our study, we compared the mean expression of each pair of homeologs calculated using ~200 accessions. This comparison indicates that the vast majority of homeologs have equal population-scale expression means (essentially 1:1 expression), and the usage of the term “balanced expression” seems appropriate here. Besides, we did not state that these homeologs are balanced. We concluded that strong positive correlation indicates that there are a lot of accessions in population with balanced homeologs. To avoid confusion with the previously used terminology, we have modified this sentence to replace “balanced homeologs” with “homeologs with matching expression” as follows: “While a strong positive correlation would be indicative of matching homeolog expression levels (Fig. 1a) in most accessions in the panel, an increase in the proportion of accessions with biased homeologs would decrease SCC (Fig. 1a).”

We also did not observed in our data cases provided by the reviewer, where expression follows the 4:1 ratio across all accessions.

Comment 3: The term “dysregulated” might be more accurately described as “negatively correlated homeologs”. This is what is described in the methods (line 393-942) which says a dysregulated homeolog should have a negative correlation ($SCC < 0$) with two homeologs, and a strong negative correlation ($SCC < -0.4$) with at least 1 homeolog. However, the Figure 1

legend describes “dysregulated homoeologs show different levels of expression” which to me means something quite different (i.e. A expression is higher than B expression). The explanation of dysregulated homoeologs should be made clearer and consistent throughout the paper.

Response: Please see response to comment 6 from reviewer 3 and response to previous comment 2 from this reviewer.

During writing the manuscript, we also felt that the usage of “homeolog dysregulation” term might be confusing and also complicate connecting our findings to existing body of work on regulation of polyploid gene expression. In the revised manuscript, we decided to stick with previously used terminology that appear to be sufficient to describe our results. To describe unequal expression of homoeologs at the individual accession level, we used “homoeolog expression bias” or “homoeologs with biased expression”. We used “negatively correlated homoeologs” term to describe cases of homeologous gene pairs where increased frequency of accessions with biased gene expression leads to negative expression correlation in the panel.

Comment 4: Similarly, the definition and consistency of the 59 homoeologs identified as being dysregulated could be made clearer. Was the level of dysregulation and homoeolog expression exactly the same between seedlings and spikes for each line? More generally, how consistent are dysregulated homoeologs across tissue, time and biological replicate? It not evident that expression at this timepoint being studied is the only one affecting trait expression. If homoeolog contribution changes with development, it is possible other homoeologs are identified as dysregulated at a timepoint or tissue not investigated here and may have a more direct impact on trait values. Ideally, the tissues underlying the basis for a given trait should be studied (e.g. grain gene expression for a grain related trait) but this study provides a first pass attempt at trying to identify associations. This limitation should be discussed, and support for the approach used comes from work in maize which shows that several different tissues can be used to predict seed-weight (Kremling et al., 2018).

Kremling, K., Chen, SY., Su, MH. et al. Dysregulation of expression correlates with rare-allele burden and fitness loss in maize. *Nature*. 2018. 555, 520–523

Response: As was described in the response to the previous comment, we use term “negatively correlated homoeologs” to define these 59 homoeologs.

Thank for pointing out at these aspects of analyses, which were not well presented in the previous version of the manuscript. Overall, we observed good consistency in the levels of homoeolog expression correlation across two tissues analyzed in our study (seedlings and spikes). This was indicated in the Results, section “Population-scale homoeologous gene expression variation”: “*The SCCs calculated for the same sets of homoeologs using RNA-seq data from both the seedlings and spike tissues²⁸ collected from a distinct set of accessions were generally similar, suggesting that tissue-specific factors do not substantial affect co-expression of the majority of homoeologs at the population scale (Supplementary Fig. 1c).*”

On average, the same set of homoeologs in the spikes also showed the lack of coordinated expression, although not as severe as in the seedlings. We added this statement into the Results, section “*Joint eQTL and GWAS analysis detects genes ...*”: “*In the RNA-seq data from spikes, these homoeologs also showed the lack of coordinated expression (mean SCC = 0.03 ± 0.05), although not as substantial as in the seedlings.*”

As was suggested by the reviewer, we have also tested correlation between the trait variation and dysregulated homoeologs identified using similar strategy in the spikes. In total, we have identified 67 homoeologs showing evidence of dysregulation (negatively correlated). We used matching phenotyping data published in the same study (Wang et al.

2017, 10.1104/pp.17.00694) reporting the RNA-seq data from spikes. We found that the SCC between the seed number and number of spikelets per spike also show negative correlation with the number of dysregulated genes (SCC = -0.25 and SCC = -0.16, respectively). These correlations are lower than SCC = -0.35 observed between dysregulated homoeologs in seedlings and number of spikelets per spike.

We have added these results into the Results section: “Similar analysis performed using the negatively correlated homoeologs detected in the spikes and the number of grains and the number of spikelets per spike also revealed negative correlation between the low-expressing homoeologous alleles and traits (SCC = -0.25 and SCC = -0.16, respectively).”

Overall, comparison of relationship between trait expression and homoeolog dysregulation across tissues suggests that dysregulated homoeologs detected in both seedlings and spikes correlate with the trait values.

Comment 5: The conclusion (line 243-245) that “our results suggest that the dysregulation of homoeologs (Fig. 1a) is primarily associated with the cis-regulatory diversity” is not fully supported by the evidence presented.

The majority of the results do support that cis-genomic diversity is associated with homoeolog dysregulation, but this does not go as far as supporting “cis-regulatory diversity” because the analysis was done at a subgenome scale (e.g. A vs B vs D), rather than at a specific chromosome scale.

Therefore, I find the current statement misleading because cis-regulatory would normally refer to a region close to the gene being regulated (or at the least on the same chromosome) but in this case most of the data shows the regulation comes from the same subgenome, but not necessarily the same chromosome.

A way to improve this analysis, could be to re-run the partitioning of genetic variation per chromosome, rather than per subgenome, then the data could support this conclusion. Otherwise, the conclusion should be amended to “our results suggest that the dysregulation of homoeologs (Fig. 1a) is primarily associated with the cis-genomic diversity”.

Response: We have performed additional analyses presented on Fig. 2c now, which show that our conclusion was largely correct and that main difference between homoeologs that show high and low levels of expression correlation is the proportion of variance explained by *cis*-variants near genes rather than *trans*-variants.

Comment 6: Three comments related to the assessment of purifying selection homoeologs and singletons in Fig 4d:

a) There is no explicit test for purifying selection in Fig 4d. One way to do this, as done for many of the other analyses in this paper, is to permute the expression and genotype, separately for each MAF category, and establish the significant threshold.

b) Related to the same analyses, the effect size for eQTLs with low MAF can be inflated for a number of reasons and detecting rare small effect QTLs in general is difficult. It would be worth repeating the analyses using a fixed contribution of two alleles by subsampling genotypes for the more common allele and repeating the effect size calculations. The results of Fig S2C suggests this is unlikely to alter conclusions regarding the comparison between singletons and homoeologs.

c) Why not consider doublets and triplets separately?

Response: The main goal of this analysis was to test whether selection is relaxed in duplicated genes compared to singletons or not. This comparison of the effect size versus allele frequency provides simple approach to test this hypothesis, and was used previously in other systems (plant example DOI: 10.1073/pnas.1503027112; human example DOI: 10.1534/genetics.118.301833). There are a number of other studies that investigated the mode of selection acting on gene expression and demonstrated that expression is under purifying selection.

We do agree with the reviewer that effect size of alleles with low frequency could be inflated. This was also the concern raised by another reviewer. To address this issue, we have performed analyses using a subset of eQTL whose effects could be detected at all frequencies in population (see response to the comment 8 from Reviewer 2). Using this subset of eQTL, we show that negative relationship between allele frequency and effect size in homoeologous gene triplets still persists. We have updated Results section and added Supplementary Fig. 7 to incorporate these analyses into the manuscript.

“The negative relationship between frequency and effect size was observed in homoeologs even for the subset of *cis*-eQTL whose effects are detectable at all frequencies (Supplementary Fig. 7)”.

Unfortunately, the number of duplets with detected eQTL is not sufficiently large to perform this test using duplets as a separate group.

Methodological major comments:

Comment 7: It would be important to know if the key conclusions still hold when excluding genes that show high levels of expression noise – variability between biological replicates. Since

the three samples together were ground together, this current data does not allow one to assess this.

Response: The pooling strategy has been used in many studies aimed at eQTL mapping or studying effect of selection of gene expression, for example, including maize (1 - Kremling et al. Dysregulation of expression correlates with rare-allele burden and fitness loss in maize. *Nature*; 2018;555:520–3; 2 - Liu S et al. *Genome Biol.* 2020;21:1–22) and *C. grandiflora* (Josephs et al. *Proc Natl Acad Sci U S A.* 2015;112:15390–5). In our study, we used two RNA-seq datasets derived from different sets of accessions and different tissues. One RNA-seq dataset was derived from wheat seedlings, another RNA-seq dataset was taken from the previously published study (www.plantphysiol.org/cgi/doi/10.1104/pp.17.00694). Conclusions derived in the paper hold in both datasets, and in most cases consistent with the expectations or conclusions reached in other studies. Considering these results, and the fact that many conclusions reached in our study are based on methods and analyses previously also applied to datasets generated for pooled RNA, make us believe that pooling had little impact on the results of our analysis. The biological replicate pooling could affect comparison of gene expression between a pair of individuals from population. Indeed, as reviewer pointed, in this case, one could lose the ability to take into account the effect of genes showing extreme levels of variability in expression. However, in our eQTL mapping study, we report associations detected using SNPs with $MAF \geq 0.05$ (out of 200 accessions). This means that any of these mapped eQTL is based on association tests conducted between a genetic variant and expression values from at least 10 accessions. In other words, any expression statistics derived in our study is based on at least 10 independent expression values (200 accessions * 0.05 MAF = 10). And even if each value is based on pooled sample, combined together they should provide quite accurate estimate of expression differences for any gene between the groups of individuals combined based on the genotype at eQTL locus.

Comment 8: Is Hi-C stable between lines? Hi-C data from CS is being used to infer relationships in all lines, yet they are very different genotypes which are likely to have large structural re-arrangements. Therefore, whilst interesting, these results are a bit speculative. How robust are comparisons when Hi-C datasets from different genotypes are used? One way to test this would be use the datasets generated Walkowiak et al. (2020) and test if the results hold under all conditions.

Walkowiak S, Gao L, Monat C, Haberer G, Kassa MT, Brinton J, Ramirez-Gonzalez RH, Kolodziej MC, Delorean E, Thambugala D, Klymiuk V. Multiple wheat genomes reveal global variation in modern breeding. *Nature*. 2020 Nov 25:1-7.

Response: There are some conclusions in the manuscript based on Hi-C analyses could be affected by structural variation among accessions. Based on the comments from other reviewers, we decided to retain in the manuscript only enrichment analyses of eQTL in the regions on chromatin contacts. The analysis of eQTL and Hi-C contacts is an attractive direction of further research, but it is outside of the scope of our already quite extensive analyses performed in our study (please see responses to comments 1, 3 and 4 from Reviewer 1). Following advice from other reviewers, we have shortened manuscript to provide more focused and concise story. Besides, since these enrichment analyses are based on genome-wide data and seek to show over-representation of eQTL in the regions of high contact frequency, there are less affected by local

structural variation. The Results section and Fig. 3 (the Fig. 5 was removed and parts of this figure were moved to Fig. 3) have been modified to reflect these changes.

Comment 9: Line 117-119 “Transcripts Per Million (TPM) were estimated for high-confidence (HC) genes in RefSeq v.1.0, with 82,092 genes (66,333 genes) showing TPM > 0.5 in at least two wheat lines”. But this does not agree with what is stated in the methods (Lines 701-704) which states that “Gene models with expression standard deviation > 0.5 and expressed (TPM > 0.5) in at least three samples have been used in our analyses. This set of genes included 52,511 HC gene models, 29,226 LC gene models, and 13,861 de novo assembled transcripts.” Were only HC genes used, or were LC and de novo transcripts also included?

Response: Thank you for noticing this inconsistency. These numbers are incorrect and came from earlier analyses performed using the StringTie program. The Methods section provides correct information based on the calculations of gene expression using Kallisto. We have updated this section. Also, because all our analyses are based on gene models predicted in the reference genome, we have removed information about de novo transcripts. In all further analyses, only HC gene models were used “... were estimated for high-confidence (HC) genes in RefSeq v.1.0, with 52,511 transcripts (47,274 genes) showing TPM > 0.5 in at least three wheat lines...”.

Comment 10: Line 700. How were the de novo transcripts assembled? And which analyses were they used for?

Response: De novo assembled transcript have not been used in this manuscript. The description of de novo transcriptome assembly was removed.

Comment 11: Data availability:

Deposited RNA-seq data is private until 1st Nov 2023, this must be made public upon publication. Also, this is not the raw RNA-seq reads (is this in the linked SRA record? I cannot access it to check). The raw fastq files should also be made available upon publication. Finally, these SRA and GEO accession numbers should be included in the main methods or in list of supplemental data so that others can find them.

Response: Both RNA-seq and TMP data were deposited to NCBI and will be released along with the published manuscript. The reviewers could get access to data using information that was provided to the Nature Communications during the submission (see below).

SRA:

<https://dataview.ncbi.nlm.nih.gov/object/PRJNA670223?reviewer=o5h7sh4bb3j7ntmguk02as41hg>

GEO: <https://www.ncbi.nlm.nih.gov/geo/query/acc.cgi?acc=GSE167479>

Enter GEO token into the box: gfwveaumvbghlwb

Minor comments:

Comment 12: Title is very long. Could it be re-written to be shorter and more informative?

Response: We tried to shorten title and make it more informative. The current title is “The genetic architecture of homoeologous gene expression dosage variation and its impact on agronomic traits in allopolyploid wheat”.

Comment 13: Are the 2Mb regions defined surrounding gene always intergenic regions? If closer regions (500 bp and 1 MB) are less likely to intersect with neighbouring genic regions, they could be used instead.

Response: We used 2 Mb interval just to define cis-eQTL – a variant most strongly associated with the expression of a target gene. The selection of this distance threshold is based on the fact that wheat has large genome and the rate of LD decay is slow across most of the genome (50% decay within 5-10 Mb, see for example: <https://www.ncbi.nlm.nih.gov/pmc/articles/PMC5701291/>). Both 1 Mb and 0.5 Mb intervals are too short for appreciable decrease in LD for wheat. Within the 2 Mb interval, we selected most significant SNP. In most cases, these most significant cis-eQTL were much closer to a gene (~1-10 kb) than 2Mb, as could be seen from the eQTL density distribution in Supplementary Fig. 6b, suggesting relatively small overlap with the neighboring genic regions.

Comment 14: Text on figure 1 is too small to read. Figure 1B does not really add to the narrative and could be removed.

Response: Fig 1. was modified as suggested. We used font size 6, which is allowed by the publisher. In case if manuscript will be published, we believe that text will be readable.

Comment 15: In Fig 3A and B, what does deleterious mean? Missense can also be deleterious.

Response: We used conservative criteria to define deleterious as those mutations that lead to premature termination codons due to nonsense and splice-site disruption mutations annotated using SNPEffect program. For clarification, we have added the following sentence in the methods: “SNPs resulting in splice-site disruption and premature termination codons were considered as putatively deleterious.”

Comment 16: Figure 5, text too small panel e). Explanation of tpm graphs on the right not clear. Also why is this panel inside a blue box?

Response: Fig. 5 has been modified in response to other reviewers’ comments and parts of the figure 5 under question are removed. The remaining parts of former Fig. 5 are now included into Fig. 3.

Comment 17: Figure 7C, the Y axis is misleading, it's a count of low expression alleles for dysregulated homoeologs in each line, or so I think.

Response: Thank you for pointing at this part of the figure. Yes, this is the count of low-expressing alleles. This figure with modified legend is now part of Fig. 6c.

Comment 18: Is Line 560 supposed to say Fig 7F?

Response: This part of the figure was removed.

Comment 19: Line 593 “Previous studies in maize” – only 1 study is cited.

Response: This typo was corrected

Comment 20: Line 683. Which programme and parameters were used for read mapping? Which genome sequence was used as a reference?

Response: The requested details are added into the Methods.

Comment 21: Line 784 “All SNP sites with missing rate > 75% or heterozygosity rate < 3% were removed”. Why remove SNPs with low heterozygosity? I would expect most wheat lines to be homozygous. (This is also the opposite of the stated filter on line 779)

Response: Thank you for noticing this typo, which was corrected. It should be “>3%”.

Comment 22: Line 926 to 928 there is a sentence repeated “If the absolute....”

Response: Corrected.

Comment 23: Supplementary table 2, the column named “GENE_ID (v.2.0)” should be called “Gene_ID (v1.1)”, consistent with IWGSC et al., 2018.

Response: Corrected

Comment 24: Supplementary table 12 is labelled as Supplementary Table 13 in the excel file so needs to be corrected.

Response: Corrected.

Comment 25: Supplemental Fig 14. What are eGenes?

Response: The term eGene was initially used in the manuscript to describe a gene that is associated with eQTL. The eGene is replaced by “target gene” across the manuscript.

Reviewers' Comments:

Reviewer #1:

Remarks to the Author:

The authors have addressed most of my concerns and improved the manuscript.

I still have one point which mostly involves text editing.

Authors wrote in the main text line 293 “The *cis*- and *trans*-eQTL showed similar levels of enrichment across the various epigenetic marks, with both types of eQTL enriched for epigenetic modifications associated with enhancers (H3K4me1), transcription (H3K36me3), activation or regulation of gene expression (H3K27ac, H3K4me3).

Authors also wrote in their Response to the Reviewers’ Comments

“The H3K4me3, H3K4me1, H3K27ac marks are well known for being associated with enhancers and promoters of actively expressed genes. [...] 1) “H3K4me1 is highly enriched at enhancers, ... and H3K4me3 is a hallmark of the promoters of actively transcribing and poised genes” (DOI: <https://doi.org/10.1038/emm.2017.11>); 2) “Selected epigenomes also contain a subset of additional epigenomic marks, including: acetylation marks H3K27ac and H3K9ac, associated with increased activation of enhancer and promoter regions” (DOI: [10.1038/nature14248](https://doi.org/10.1038/nature14248)); 3) “... H3K27me3 was shown to be highly enriched at the promoters of thousands of genes that are responsible for embryonic development and differentiation” (DOI: <https://doi.org/10.1038/emm.2017.11>).”

This is how it is in animal but not in plant and the paper cited refer to animals. In plant H3K4me3 and H3K27ac are mainly associated to the first nucleosome after the TSS whereas H3K4me1 is mainly associated to the gene body (see figure below for maize from (<https://doi.org/10.1038/s41467-019-10602-5> and rice from DOI: [10.1093/mp/sst018](https://doi.org/10.1093/mp/sst018))

In this context, to my view authors must edit the text since most of the data proved that in plants neither H3K4me1 nor H3K27ac marked enhancers. (see also <https://doi.org/10.1038/s41467-019-09513-2> in this paper it is clearly demonstrated that H3K27ac is not a hallmark of enhancers in Arabidopsis.)

Reviewer #2:

Remarks to the Author:

I thank the authors for the additional work in response to my previous comments. Most of my earlier concerns have been addressed. However, I have a few additional comments.

Comment 1

PEER factors are often used to account for structure in gene expression data (e.g., systematic differences in gene expression among subsets of individuals), and principal components (PCs) derived from genetic data are often used to account for population stratification (i.e., different subsets of the sample come from different populations). Genetic PCs are effective in accounting for population stratification but not cryptic relatedness. In the GTEx data, related donors were identified using genetic data and removed from the eQTL mapping analysis so that the GTEx eQTL results are less likely to be affected by cryptic relatedness. In this study, however, the level of relatedness is unclear. Although it is reassuring to see that the mean of GIFs across genes is close to unity, the large variability is of concern, which reflects a small effective number of independent markers, an indication of a possibly high level of relatedness.

Comment 2

Regarding my comment on the choice of significance level, the authors responded that "The false positive rate calculated using the permuted data for spikes resulted in 0.8×10^{-3} ". This value should be interpreted as is the false discovery rate (proportion of false positives in the positive results) rather than the false positive rate (proportion of false positives in all the tested results).

Reviewer #3:

Remarks to the Author:

The authors have extensively revised the manuscript which is much improved in my opinion. Below are my comments:

Throughout the manuscript, were p-values considered for SCC? Were those negative correlation significant? For example in Fig 5C, are the grey points supposed to indicate negative correlation? L502 indicated SCC = -0.18, but the plot looks weakly positive.

L195 - "only 47.8%" why say "only"? Shouldn't 47.8% indicate a higher than 1/3 expectation in D genome, in order to speculate that bottleneck decreased the diversity in the D genome more than in A and B genomes?

Fig2D and L254 - Please introduce why analyzing common and rare alleles when first mentioned, not until later in page 24.

L317-328 - This new analysis using external ATAC-seq dataset is unconvincing. Multiple steps of data manipulation was conducted to search for what the authors wanted to find, but the association between eQTL effect and promoter accessibility was not well supported. The authors' reasoning about not to include such analysis works for me, especially after the removal of Hi-C section.

L345-348 - It is not clear to me how to obtain these information from Fig 3f.

L365 should be "is supported by the finding"

How to measure "effective size" of eQTL should be explained.

Fig. 7: title still used dysregulation; d&e were not mentioned or explained in results

L622: Isn't it possible that the lack of cis-regulatory diversity is ancestral? Any evidence showing the diversity was actually lower in breadwheat than tetraploid wheat and D genome ?

Reviewer #4:

Remarks to the Author:

The authors have addressed the majority of the comments and the paper has been improved. I believe this is an interesting piece of work which adds value to the field.

Major comments:

1. In the new analysis, why were cis-genic SNPs defined as ± 1 Mb around genes (line 200) but cis-acting variant defined as those occurring within 2 Mb of a gene for the eQTL analyses (line 273)?
2. Line 334. Add a statement to make it clear that the Hi-C data is from Chinese Spring, a cultivar that was not in the eQTL panel, to aid reader interpretation (as is done for the ATAC-seq on line 319-320 which helps a lot)
3. Double check everything due to large datasets and some minor typos and errors that have come through in this revised version.

Minor comments:

1. Line 89 "To functionally characterize variants associated with homoeolog expression variation". Functional characterisation seem a bit strong because there is no direct causation shown (e.g. by gene editing). Same comment for line 282.
2. Line 120-121 makes it sound like what gene are in two distinct states- triplets or singletons. However there are many different combinations possible (e.g. 2 A copies with 1 B copy etc. as in IWGSC et al., 2018). The sentence should be updated to reflect this.
3. Figure 1a. What do the red and green colours mean in the top part of panel a) directly under "homoeologs with biased expression"?
4. Lines 161 and 164 Fig 1d is now 1c
5. Figure 3 – order of g and f is inverted
6. Figure 7- remove reference to dysregulation in figure title because it is not used in rest of manuscript

RESPONSE TO REVIEWER COMMENTS

Reviewer #1: The authors have addressed most of my concerns and improved the manuscript.

Comment 1: I still have one point which mostly involves text editing. ... In plant H3K4me3 and H3K27ac are mainly associated to the first nucleosome after the TSS whereas H3K4me1 is mainly associated to the gene body (see figure below for maize from (<https://doi.org/10.1038/s41467-019-10602-5> and rice from DOI: 10.1093/mp/sst018). In this context, to my view authors must edit the text since most of the data proved that in plants neither H3K4me1 nor H3K27ac marked enhancers. (see also <https://doi.org/10.1038/s41467-019-09513-2> in this paper it is clearly demonstrated that H3K27ac is not a hallmark of enhancers in Arabidopsis.)

Response: Thank you for pointing out at the differences in the enrichment of epigenetic marks in plant and mammalian genomes, and providing relevant references. We cited those references that report maize and rice data, as these monocot species are more closely related to wheat than Arabidopsis (dicot). We corrected a senescence in Results: "...with both types of eQTL enriched for epigenetic modifications associated with gene body (H3K4me1), transcription (H3K36me3), and active expression (H3K27ac, H3K4me3) (Figs. 3e, 3f)[refs: 27, 38, 39]."

Reviewer #2: I thank the authors for the additional work in response to my previous comments. Most of my earlier concerns have been addressed. However, I have a few additional comments.

Comment 1: PEER factors are often used to account for structure in gene expression data (e.g., systematic differences in gene expression among subsets of individuals), and principal components (PCs) derived from genetic data are often used to account for population stratification (i.e., different subsets of the sample come from different populations). Genetic PCs are effective in accounting for population stratification but not cryptic relatedness. In the GTEx data, related donors were identified using genetic data and removed from the eQTL mapping analysis so that the GTEx eQTL results are less likely to be affected by cryptic relatedness. In this study, however, the level of relatedness is unclear. Although it is reassuring to see that the mean of GIFs across genes is close to unity, the large variability is of concern, which reflects a small effective number of independent markers, an indication of a possibly high level of relatedness.

Response: It is quite unlikely that our sample carries significant number of highly related accessions. The selection procedure applied in our study would rather lead to extremely diverse sample of accessions rather than accessions sharing significant recent ancestry. We should apologize for not providing more detailed information related to sample selection, which is now added to the Methods section. Briefly, our panel was selected from a worldwide sample including more than 2,000 lines previously genotyped using the 9K iSelect SNP assay (Cavanagh et al., 2013) and having previously collected data on resistance to multiple races of stem rust. During selection we tried to maximize: 1) genetic diversity, 2) representation of diverse geographic regions, and 3) representation of phenotypic response to distinct strains of fungal pathogen. We believe that this should provide adequate protection against choosing highly related individuals. The low false-discovery rate estimated by permutation supports this conclusion indicating that the high-level of line relatedness unlikely should be a major issue in our mapping study.

Comment 2: Regarding my comment on the choice of significance level, the authors responded that "The false positive rate calculated using the permuted data for spikes resulted in 0.8×10^{-3} ". This value should be interpreted as is the false discovery rate (proportion of false positives in the positive results) rather than the false positive rate (proportion of false positives in all the tested results).

Response: Thank you for pointing out at this error. Two sentences describing the permutation results in Methods have been corrected.

Reviewer #3: The authors have extensively revised the manuscript which is much improved in my opinion. Below are my comments:

Comment 1: Throughout the manuscript, were p-values considered for SCC? Were those negative correlation significant? For example in Fig 5C, are the grey points supposed to indicate negative correlation? L502 indicated $SCC = -0.18$, but the plot looks weakly positive.

Response: P-values were calculated for negatively correlated homoeologs using expression values from all 198 accessions, and only those that are significant were reported. In Fig. 5c, p-value based on 198 pairs of expression values was significant (p-value = 0.01, now added to the text). The green and yellow data points show expression values for wheat lines without or with terminal deletion on chr. 1D, respectively.

Comment 2: L195 - "only 47.8%" why say "only"? Shouldn't 47.8% indicate a higher than 1/3 expectation in D genome, in order to speculate that bottleneck decreased the diversity in the D genome more than in A and B genomes?

Response: Thank you for the suggestion. We have modified this sentence.

Comment 3: Fig2D and L254 - Please introduce why analyzing common and rare alleles when first mentioned, not until Later in page24.

Response: Thank you for the suggestion. We have added introductory sentences in the Results section: "The effects of cis- and trans-acting variants on expression correlation between homoeologs".

Comment 4: L317-328 - This new analysis using external ATAC-seq dataset is unconvincing. Multiple steps of data manipulation was conducted to search for what the authors wanted to find, but the association between eQTL effect and promoter accessibility was not well supported. The authors' reasoning about not to include such analysis works for me, especially after the removal of Hi-C section.

Response: We do agree with the reviewer, and decided to exclude this analysis from the manuscript.

Comment 5: L345-348 - It is not clear to me how to obtain these information from Fig 3f.

Response: This information could be obtained from Supplementary Table 7, which was cited in the same section of Results. We have added the reference to this table.

Comment 6: L365 should be "is supported by the finding"

Response: Corrected.

Comment 7: How to measure "effective size" of eQTL should be explained.

Response: The effect size of eQTL is a linear regression slope estimated by Matrix eQTL for each variant tested. We have added this clarification to the Methods section.

Comment 8: Fig. 7: title still used dysregulation; d&e were not mentioned or explained in results.

Response: Thank you for noticing this error. Also, we mistakenly kept this figure labeled as Fig. 7, whereas it is Fig. 6. The title is corrected: "Figure 6. Biased expression of homoeologous genes is linked with variation in productivity traits." Both panels "6d" and "6e" have already been described in the Results.

Comment 9: L622: Isn't it possible that the lack of cis-regulatory diversity is ancestral? Any evidence showing the diversity was actually lower in breadwheat than tetraploid wheat and D genome ?

Response: There is strong evidence that bread wheat experienced significant genome-wide loss of genetic diversity compared to its tetraploid and diploid ancestors. The most severe loss happened in the D genome (now it has 2-3 times lower diversity than the A and B genomes), which is consistent with nearly

50% of the trans-only regulated genes being located in the D genome, and remaining trans-only genes (~25% per genome) being located in the A and B genomes. We have cited several studies in the manuscript that specifically investigated the impact of domestication and selection on diversity in wheat. See references 60, 63, 65-67. Comparison of genetic diversity between genes regulated only by trans-eQTL with cis—only genes in wheat showed significant difference (see Fig. 2b). However, comparison of the same genes in tetraploid wild emmer using our previously published 1000 exome data (He et al, Nat. Genetics, 2019) showed no difference, suggesting that preferential trans-regulation in wheat is linked with loss of diversity during improvement. We have added this information to the results: “...no significant reduction of diversity between cis- and trans-only regulated genes (Wilcoxon rank sum test p-value = 0.1) was found in wild emmer using data from the previously published study”.

Reviewer #4: The authors have addressed the majority of the comments and the paper has been improved. I believe this is an interesting piece of work which adds value to the field.

Major comments:

Comment 1: In the new analysis, why were cis-genic SNPs defined as ± 1 Mb around genes (line 200) but cis-acting variant defined as those occurring within 2 Mb of a gene for the eQTL analyses (line 273)?

Response: In both cases we used ± 1 Mb around a gene. For consistency, we replaced “2 Mb of a gene” to “ ± 1 Mb around gene”.

Comment 2: Line 334. Add a statement to make it clear that the Hi-C data is from Chinese Spring, a cultivar that was not in the eQTL panel, to aid reader interpretation (as is done for the ATAC-seq on line 319-320 which helps a lot).

Response: The note that cultivar Chinese Spring is not part of the diversity panel is added to this sentence.

Comment 3: Double check everything due to large datasets and some minor typos and errors that have come through in this revised version.

Response: Thank you. We checked datasets and manuscript for errors and typos.

Minor comments:

Comment 1: Line 89 “To functionally characterize variants associated with homoeolog expression variation”. Functional characterisation seem a bit strong because there is no direct causation shown (e.g. by gene editing). Same comment for line 282.

Response: Thank you. We have re-worded these parts of the sentences.

Comment 2: Line 120-121 makes it sound like what gene are in two distinct states- triplets or singletons. However there are many different combinations possible (e.g. 2 A copies with 1 B copy etc. as in IWGSC et al., 2018). The sentence should be updated to reflect this.

Response: Thank you for the suggestion. We clarified that homeologs could be present in one, two or three genomes.

Comment 3: Figure 1a. What do the red and green colours mean in the top part of panel a) directly under “homoeologs with biased expression”?

Response: We have added clarification to the figure legend: “Red and green colors show low-expressing homoeologs in the A and B genomes, respectively”.

Comment 4: Lines 161 and 164 Fig 1d is now 1c

Response: Thank you! Corrected.

Comment 5: Figure 3 – order of g and f is inverted

Response: Corrected.

Comment 6: Figure 7- remove reference to dysregulation in figure title because it is not used in rest of manuscript

Response: Corrected.

Reviewers' Comments:

Reviewer #2:

None

Reviewer #3:

Remarks to the Author:

The authors have addressed my previous concerns.

Reviewer #4:

Remarks to the Author:

The authors have now addressed all my comments. Thank you.

Comment: In particular, Reviewer #2 points out that there are no direct evidences to show the degree of relatedness of the sampled accessions, especially given the large variation across transcripts and genes in terms of genomic inflation factors provided in Supplementary Figure 15. In this revision, we ask you to do the analysis by using the available SNP data to estimate the relatedness and make a histogram of off-diagonal elements of the genetic relatedness matrix.

Response: We have calculated the distribution of genetic relatedness between all possible pairs of accessions in our sample using an algorithm implemented in PLINK. This analysis (Supplementary Fig. 15) clearly shows that our panel of lines does not have highly related accessions (relatedness approaching 1.0) with nearly all pair-wise showing relatedness levels between -0.2 and 0.2. We have added a new supplementary figure showing the distribution of off-diagonal relatedness in the matrix generated by PLINK. The description of this data is added to the Methods: “The genetic relatedness analysis of this subset of lines was performed using an algorithm implemented in PLINK v.1.9. For this purpose, we have used genome-wide SNPs generated by the regulatory sequence capture and sequence-based genotyping approaches. This analysis shows that our panel does not contain highly related accessions, which otherwise might increase the chances of detecting spurious associations in GWAS (Supplementary Fig. 15).”

Reviewers' Comments:

Reviewer #2:

None